# Androgen signaling uses a writer and a reader of ADP-ribosylation to regulate protein complex assembly

Chun-Song Yang[1,2], Kasey Jividen[1], Teddy Kamata[1,2], Natalia Dworak[1], Luke Oostdyk [1,2], Bartlomiej Remlein[1,2], Yasin Pourfarjam[3], In-Kwon Kim [3], Kang-Ping Du[4], Tarek Abbas[1,2,4], Nicholas E. Sherman[5], David Wotton[1,2] & Bryce M. Paschal [1,2✉]

Androgen signaling through the androgen receptor (AR) directs gene expression in both normal and prostate cancer cells. Androgen regulates multiple aspects of the AR life cycle, including its localization and post-translational modification, but understanding how modifications are read and integrated with AR activity has been difficult. Here, we show that ADP-ribosylation regulates AR through a nuclear pathway mediated by Parp7. We show that Parp7 mono-ADP-ribosylates agonist-bound AR, and that ADP-ribosyl-cysteines within the N-terminal domain mediate recruitment of the E3 ligase Dtx3L/Parp9. Molecular recognition of ADP-ribosyl-cysteine is provided by tandem macrodomains in Parp9, and Dtx3L/Parp9 modulates expression of a subset of AR-regulated genes. Parp7, ADP-ribosylation of AR, and AR-Dtx3L/Parp9 complex assembly are inhibited by Olaparib, a compound used clinically to inhibit poly-ADP-ribosyltransferases Parp1/2. Our study reveals the components of an androgen signaling axis that uses a writer and reader of ADP-ribosylation to regulate protein-protein interactions and AR activity.

[1] Center for Cell Signaling, University of Virginia, Charlottesville, VA, USA. [2] Department of Biochemistry and Molecular Genetics, University of Virginia, Charlottesville, VA, USA. [3] Department of Chemistry, University of Cincinnati, Cincinnati, OH, USA. [4] Department of Radiation Oncology, University of Virginia, Charlottesville, VA, USA. [5] W. M. Keck Biomedical Mass Spectrometry Laboratory, University of Virginia, Charlottesville, VA, USA. ✉email: paschal@virginia.edu

The androgen receptor (AR) transduces the effects of androgen by acting as a ligand-regulated transcription factor. Androgen binds with nanomolar affinity to the C-terminal ligand-binding domain (LBD) of AR and induces structural changes within the LBD that promote intramolecular interactions with the N-terminal domain (NTD)[1–3]. Ligand-induced alterations in AR structure facilitate protein–protein interactions fundamental to AR regulation as a transcription factor[3,4]. One striking effect of androgen binding to AR is the induction of post-translational modifications. AR contains >20 sites that undergo modification, which include phosphorylation, acetylation, SUMOylation, and ubiquitination[5,6]. To date, the most abundant post-translational modification on AR is phosphorylation, which maps primarily to the NTD. Given that the NTD mediates the transactivation function of AR, androgen-regulated LBD–NTD interactions[7] could modulate AR output through changes in protein structure and post-translational modifications that specify co-factor recruitment.

Because ligand binding plays a crucial role in AR regulation, therapeutic strategies for prostate cancer have largely focused on reducing androgen synthesis and the development of anti-androgen compounds[8]. Androgen deprivation and treatment with anti-androgens also seem to mimic the genetic loss of homologous recombination (HR) genes such as BRCA1/2 and ATM which occurs in ~10% of prostate cancer patients[9]. Reduced HR activity renders solid tumors responsive to poly(ADP-ribose) polymerase (Parp) inhibitors that prevent base excision repair (BER)[10,11]. Deficits in HR and BER generate synthetic lethality because tumor cells are forced to repair DNA by error-prone, non-homologous end joining (NHEJ). These relationships have generated strong interest in Parp inhibitors, which improve outcomes in ovarian, breast, and prostate cancer patients who harbor mutations in DNA repair genes[12].

Clinically used Parp inhibitors such as Olaparib and Veliparib were developed against Parp1, the founding member of an enzyme family that uses $NAD^+$ as a co-factor for post-translational modification by ADP-ribosylation[13,14]. Parp1, Parp2, and the tankyrases, Parp5a and Parp5b, ADP-ribosylate protein substrates, but also extend the initial ADP-ribose conjugate and generate poly(ADP-ribose) chains. While the functions of poly(ADP-ribose) chains are not fully understood, these structures are known to be generated in diverse cellular contexts such as DNA repair and telomere maintenance[15,16]. Most Parp family members mediate mono-ADP-ribosylation; this occurs on a variety of amino acids, is reversible by cellular hydrolases, and is predicted to impact protein activity[17]. But with the exception of bacterial toxins, relatively little is known about how mono-ADP-ribosylation contributes to protein regulation.

Here, we describe a pathway that integrates Parp function with androgen signaling and characterize a mechanism that regulates AR output. The pathway is based on androgen induction of the mono-ADP-ribosyltransferase Parp7, which, in turn, ADP-ribosylates AR on multiple cysteine (Cys) residues. ADP-ribosylation by Parp7 results in macrodomain- (MD) mediated assembly of an AR-Dtx3L/Parp9 complex. Parp7 enzyme activity, AR ADP-ribosylation, and assembly of the AR-Dtx3L/Parp9 complex are inhibited by Olaparib. Using depletion approaches and RNA-seq, we show that the Dtx3L/Parp9 complex modulates the expression of AR-regulated genes. Our data identify an androgen-Parp signaling axis that uses an ADP-ribose writer (Parp7) and reader (Parp9) to control the assembly of a transcription factor complex.

## Results

### AR forms a complex with Dtx3L/Parp9.
Ligand-induced changes in AR protein conformation underpin the interactions that are fundamental for its transcription factor activity. To identify factors that selectively bind the agonist conformation of AR, we introduced Flag epitope-tagged wild-type (WT) AR into PC3 prostate cancer cells, treated the cells with an androgen agonist (R1881), and at multiple timepoints (0–24 h), isolated AR by immunoprecipitation (IP). SDS-PAGE and silver staining revealed the R1881-induced early release of Hsp90 from AR (0 and 2 h comparison), but also a time-dependent association of ~80-kDa proteins with AR at later timepoints (9 and 24 h; Fig. 1a). By mass spectrometry (MS), the ~80-kDa proteins were identified as Dtx3L and Parp9. These two proteins form a stable heterodimer with histone E3 ubiquitin ligase (Dtx3L) and ubiquitin mono-ADP-ribosyltransferase activities (Parp9)[18–20]. The MS results were validated by probing AR IPs for Dtx3L and Parp9, and by the reciprocal approach, Dtx3L IP, and immunoblotting for AR (Fig. 1b, c). Androgen induction of AR-Dtx3L/Parp9 complex formation was not accompanied by an increase in Dtx3L and Parp9 protein (Fig. 1b, c) or RNA expression[21]. To test whether Dtx3L and Parp9 are biochemically active when bound to AR, we performed assays that measure Dtx3L E3-mediated ubiquitylation and Parp9-mediated ADP-ribosylation in the same reaction[20]. Supplementation with recombinant Dtx3L/Parp9 and biotin-$NAD^+$ reveals ubiquitylation via a Parp9 gel shift, and ubiquitin ADP-ribosylation by Neutravidin binding to biotin-ADP-ribose conjugated to ubiquitin, respectively (Fig. 1d, lanes 3–5). The same outcomes are observed with AR IP's, but only when the cells are treated with R1881 in culture, and the assays are incubated at 30 °C (Fig. 1d, lanes 6–9). Direct IP of Dtx3L/Parp9 from untreated cells also results in the recovery of ubiquitylation and ADP-ribosylation activity, without detectable AR since no R1881 was added (Fig. 1d, lanes 10, 11). These data show that the Dtx3L/Parp9 heterodimer bound to AR has ubiquitin E3 and ADP-ribosyltransferase activities.

### Androgen induces ADP-ribosylation of AR.
Parp9 contains two copies of a protein module termed the MD, a structure that binds ADP-ribose and can target the Dtx3L/Parp9 heterodimer to DNA repair sites[22,23]. We tested whether Dtx3L/Parp9 binding to AR relies on MDs by supplementing IP reactions with ADP-ribose, reasoning this condition might dissociate Dtx3L/Parp9 from AR captured by IP. Indeed, the addition of 1 μM ADP-ribose reduced the amount of Dtx3L/Parp9 bound to AR, and virtually complete dissociation occurred in the presence of 10 μM ADP-ribose (Fig. 2a). An explanation for these results is that Dtx3L/Parp9 recognizes AR that has undergone ADP-ribosylation. To determine if AR is ADP-ribosylated, we utilized the MD from *Archaeoglobus fulgidus* (Af1521) as a biochemical probe since it shows selectivity and relatively high affinity for ADP-ribose ($K_D = 126$ nM)[24]. We immobilized GST-Af1521 on glutathione beads, performed pull-down assays with extracts from PC3-AR prostate cancer cells, and examined the bound fractions by AR immunoblotting (Fig. 2b). AR was captured on GST-Af1521 beads but only if the PC3-AR cells were treated with R1881 in culture (Fig. 2c). AR binding to Af1521 was eliminated by mutating a single amino acid (Gly42Glu) in the ADP-ribose binding pocket[25]. Endogenous AR from VCaP cell extracts also bound to GST-Af1521 beads, was dependent on R1881 pre-treatment of the cells, and was competitively inhibited by ADP-ribose (Fig. 2d). All of these data are consistent with modification of AR by ADP-ribosylation in response to R1881 treatment.

### An agonist conformation is required for AR ADP-ribosylation.
Induction of AR ADP-ribosylation by R1881, a commonly used synthetic androgen agonist that induces AR conformational changes, led us to test a panel of ligands, including androgen agonists and antagonists. We found that AR ADP-ribosylation

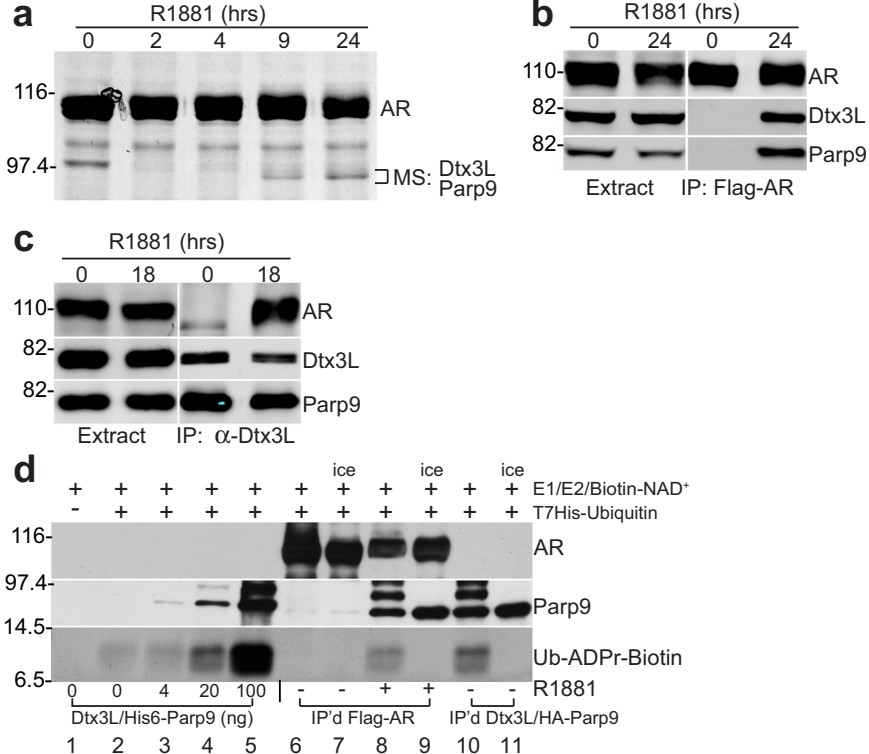

**Fig. 1 Androgen induces assembly of AR-Dtx3L/Parp9 complexes. a** PC3-AR cells were treated with the synthetic androgen R1881 for the indicated times. AR was isolated by IP, and the protein complexes were examined by SDS-PAGE and silver staining. The proteins migrating slightly faster than Hsp90 were identified as Dtx3L and Parp9 by mass spectrometry (MS). **b** AR IP's from R1881-treated PC3-AR cells immunoblotted for AR, Dtx3L, and Parp9. **c** Dtx3L IP's from R1881-treated PC3-AR cells immunoblotted for AR, Dtx3L, and Parp9. **d** Biochemical assays for ubiquitin E3 and ADP-ribosyltransferase activities in AR-Dtx3L/Parp9 complexes. Lanes 1–5 contain negative and positive controls (buffer, recombinant Dtx3L/Parp9, E1, E2, biotin-NAD$^+$) for detection of Ub-modified Parp9 and ADP-ribosylated ubiquitin. Lanes 6–9 contain AR IP's from control and R1881-treated cells incubated at 30 °C, or on ice to block the reaction. Lane 8 shows that Dtx3L/Parp9 bound to AR has E3 activity (Ub-induced mobility shift of Parp9) and mono-ADP-ribosyltransferase activity towards Ub detected via ADPr-biotin. Lane 10 shows the levels of E3 and ADP-ribosyltransferase activity detected in the Dtx3L/Parp9 directly isolated by IP. Source data are provided as a Source data file.

was induced by dihydrotestosterone (DHT) and androstenedione (ASD) but neither by the adrenal androgen dehydroepiandrosterone (DHEA), estradiol nor by various anti-androgens (Fig. 2e). The level of AR ADP-ribosylation induced by R1881 was reduced by co-treating cells with the nonsteroidal anti-androgen, enzalutamide (Enza; Fig. 2e). Deletion of the LBD abrogated R1881 induction of AR ADP-ribosylation (Fig. 2f). Moreover, a point mutation in helix 8 of the LBD (F805S) associated with complete androgen insensitivity syndrome[26] greatly reduced ADP-ribosylation and Dtx3L/Parp9 recruitment (Fig. 2g). These data, together with the ligand selectivity for ADP-ribosylation, indicate that ligand-induced changes in AR structure are critical for ADP-ribosylation to occur.

**ADP-ribosylation of AR occurs on multiple cysteines.** To identify the androgen-induced ADP-ribosylation sites, we IP'd AR from R1881-treated cells and used LC-MS/MS to identify peptides with a mass addition indicative of ADP-ribose (+541 Da). We also used the approach of treating IP'd AR with the pyrophosphatase NUDT16 since converting ADP-ribose to phosphoribosyl enables enrichment of the peptide by metal affinity chromatography[27]. In this case, the mass addition (+212 Da) reflects the phosphoribosyl group. Using these two approaches, we identified a total of 11 AR ADP-ribosylation sites, including seven sites in the NTD (Supplementary Figs. 1–13 and Supplementary Table 1). The b and y ion series generated by collision-induced dissociation (CID) unambiguously identified Cys as the ADP-ribosylated amino acid at nine

of the sites (Supplementary Figs. 1, 2, 4–10, 12). The partial b and y ions derived from the other two sites were indicative of Cys327 or Ser328, and Cys620 or Tyr621, as the ADP-ribosylated residues (Supplementary Figs. 3 and 11). Cys327 and Cys620 in these peptides did not contain carbamidomethyl, which is consistent with Cys-ADP-ribosylation blocking alkylation by iodoacetamide (Supplementary Table 1). For an independent assessment of which amino acids in AR are ADP-ribosylated, we employed a chemical sensitivity approach that relies on the cleavage of the ADP-ribose-protein linkage in vitro[28]. We found that ~99% of the covalently bound ADP-ribose was released from purified AR by exposure to the Cys-directed reagent mercury chloride (HgCl$_2$), while parallel incubations with chemicals that release ADP-ribose from the side chains of other ADP-ribose acceptors (Glu, Asp, Lys)[28] had no effect (Fig. 2h). HgCl$_2$ treatment had a negligible effect on ADP-ribose conjugated to a Parp1 fragment[29], which was released quantitatively by the Ser-ADP-ribose-directed hydrolase ARH3[30] (Fig. 2i). Lastly, HgCl$_2$ addition to cell extract from R1881-treated cells eliminated AR binding to Af1521 in pull-down experiments (Fig. 2j). The biochemical and MS/MS data show that AR undergoes androgen-dependent, Cys-ADP-ribosylation on multiple residues, although it remains possible that non-Cys residues in AR were ADP-ribosylated and went undetected by Af1521 and MS/MS.

**ADP-ribosylation of AR is sensitive to cycloheximide.** We performed an androgen treatment time course and Af1521 pull-down, and found that AR ADP-ribosylation is detected after 4 h

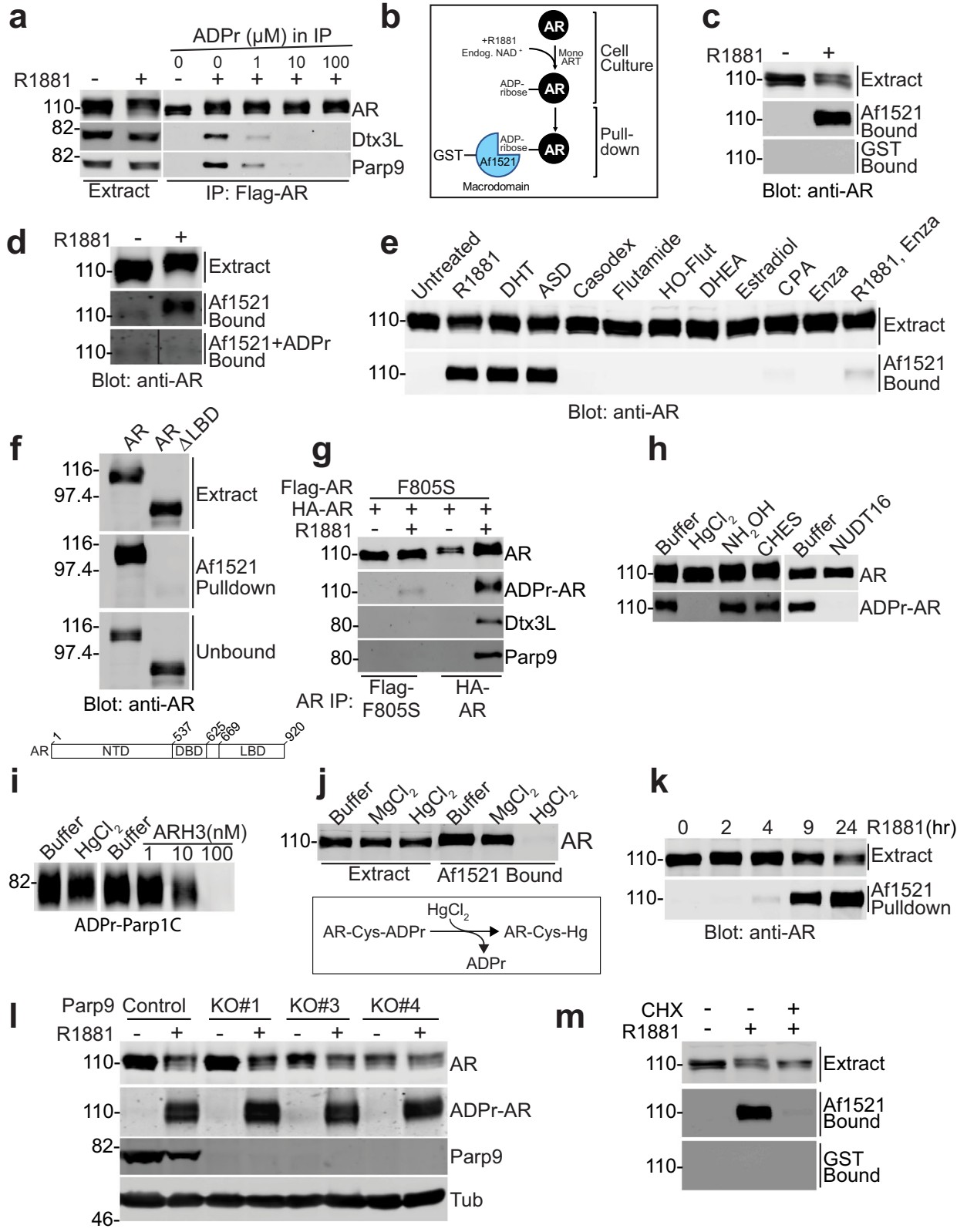

of R1881 treatment, with higher levels of ADP-ribosylation observed at 9 and 24 hr (Fig. 2k). Because the kinetics of AR ADP-ribosylation (Fig. 2k) was similar to Dtx3L/Parp9 recruitment to AR (Fig. 1a), we addressed whether Parp9 is responsible for AR ADP-ribosylation. Three independent Parp9 KO lines

generated using CRISPR displayed robust, androgen induction of AR ADP-ribosylation (Fig. 2l). Thus, Parp9 is not the enzyme that ADP-ribosylates AR in response to androgen. We found, however, that androgen induction of AR ADP-ribosylation is cycloheximide-sensitive (Fig. 2m). The requirement for new

**Fig. 2 Ligand-selective AR ADP-ribosylation occurs on cysteines. a** Immunoblot of IP'd AR, Dtx3L, and Parp9 showing the ADPr-induced dissociation of Dtx3L/Parp9 from AR. **b** Scheme for capturing ADP-ribosylated AR using GST-Af1521. **c** AR binding to Af1521 in vitro depends on androgen treatment of PC3-AR cells. Cell extracts from PC3-AR cells (untreated, +R1881) were combined with GST-Af1521 and GST beads (4 h), and bound fractions analyzed by immunoblotting. **d** AR from VCaP cells undergoes androgen-dependent binding to Af1521 beads. Extracts from VCaP cells (untreated, +R1881) were combined with GST-Af1521 beads +/− ADPr, and bound fractions examined by immunoblotting. **e** Ligand induction of AR binding to Af1521 is selective for androgen agonists. Cell extracts from PC3-AR cells treated with indicated compounds (see "Methods" for concentrations) were combined with Af1521 beads and the bound fractions immunoblotted. **f** WT AR and ΔLBD AR (residue 1–710) were expressed in cells, isolated by pulldown on Af1521, detected using AR antibodies. **g** A point mutation in the AR LBD prevents ADP-ribosylation. A stable cell line expressing both HA-tagged WT AR and Flag-tagged F805S (LBD mutant) was treated +R1881 and IP's analyzed for ADP-ribosylation, Dtx3L and Parp9. **h** Chemical sensitivity of ADPr-AR linkage. IP'd ADP-ribosylated AR was incubated with the indicated reagents (see "Methods") and the level of remaining ADP-ribose detected by Af1521. NUDT16 treatment (3 μM) quantitatively cleaved ADP-ribose from AR and eliminated the ADPr signal. **i** $HgCl_2$ has minimal activity towards Ser-conjugated ADPr. A Parp1 fragment was ADPr-modified in vitro and tested for its susceptibility to $HgCl_2$. Ser-specific linkage of ADPr was confirmed by removal with ARH3. **j** $HgCl_2$ treatment of cell extract from PC3-AR cells treated with R1881 abolishes AR binding to Af1521 beads. **k** Androgen induction of AR ADP-ribosylation is time-dependent. PC3-AR cells were treated with R1881 for the indicated timepoints, and extracts were used for Af1521 pulldown and AR immunoblotting. **l** Parp9 is not required for androgen induction of AR ADP-ribosylation. Independent clones of Parp9 knockout cells (PC3-AR cells; see "Methods") were treated +R1881 and analyzed for AR ADP-ribosylation, and for AR and Parp9 by immunoblotting. **m** Androgen induction of ADP ribosylation requires new protein synthesis. PC3-AR cells were treated without and with R1881 + cycloheximide (CHX), cell extracts used for pulldown on Af1521, and bound fractions tested by immunoblotting. Source data are provided as a Source data file.

protein synthesis suggests that AR regulates the expression of a factor required for AR ADP-ribosylation.

**The androgen-regulated, cycloheximide-sensitive factor is Parp7.** We analyzed RNA-seq data from VCaP and PC3-AR cells treated +R1881 to determine if the expression of any Parp family member is regulated by androgen. While small androgen-induced changes (up and down) were apparent for a few Parps, the only Parp whose expression increased >2-fold in both prostate cancer lines was the mono-ADP-ribosyltransferase Tiparp (Parp7; Fig. 3a). In publically available ChIP-seq data[31,32], it is clear that AR undergoes robust androgen-induced binding to the Parp7 gene, which we verified by ChIP-PCR (Fig. 3b, c). To test whether Parp7 is the factor responsible for AR ADP-ribosylation and AR-Dtx3L/Parp9 complex formation, we generated prostate cell lines for stable depletion of endogenous Parp7, and for ectopic expression of HA-Parp7. PC3-AR cells with reduced levels of Parp7 (shParp7) showed a clear defect in R1881-induced AR ADP-ribosylation and R1881-induced complex formation (Fig. 3d, e). Ectopic expression of HA-Parp7 shortened the latency and increased the levels of R1881-induced complex formation (Fig. 3f), though it did not eliminate the androgen requirement. Comparable results were obtained in VCaP cells, where stable depletion of Parp7 reduced AR ADP-ribosylation and complex formation (Fig. 3g). In a time course of R1881 treatment, there is a rapid induction of Parp7 mRNA detectable by RT-qPCR, which is followed by Parp7 protein accumulation and AR ADP-ribosylation (Fig. 3h, i). We tested for the sufficiency of Parp7 to ADP-ribosylate purified AR in vitro with enzyme prepared in insect cells. Recombinant Parp7 ADP-ribosylated AR and the modification were reversible with $HgCl_2$ (Fig. 3j).

The previous work[33] showed that Parp7 is ADP-ribosylated on Cys39. We performed MS/MS on Parp7 expressed in insect cells and found two additional ADP-ribosylation sites, Cys100 and Cys439 (Supplementary Figs. 14 and 15). Thus, Parp7 has at least three Cys-ADP-ribosylation sites, which could be the result of automodification. We used Seq2Logo[34] to ask if there are sequence features associated with Cys-directed ADP-ribosylation sites in AR and Parp7 (Supplementary Fig. 16). In AR, 6 of the 11 ADP-ribosylation sites contain Pro at the −4 or +2 position (relative to Cys), and a similar arrangement was found in 2 of 3 ADP-ribosylation sites in Parp7. Although this is a small sampling, the data suggest that Parp7 recognizes sequences that are distinct from the motifs used by Parp1 (Lys–Ser and

Ser–Gly–Gly)[35] and it prefers a neighboring Pro residue though the spacing appears different from Glu ADP-ribosylation[36]. ADP-ribosylation of AR in vitro was eliminated by deletion of the AR NTD (Fig. 3k). This result was expected since the NTD contains most of the ADP-ribosylation sites, including five sites with Pro at the −4 or +2 position. The AR NTD also contains multiple androgen-induced phosphorylation sites[6], but none of these are in close proximity to the Cys-ADP-ribosylation sites. Mutation of the five AR phosphorylation sites in the NTD (Ser83, 96, 258, 310, 426 to Ala) did not impair ADP-ribosylation or binding to Dtx3L/Parp9, and inhibiting ADP-ribosylation did not prevent phosphorylation assessed with phospho-site-specific antibodies (Supplementary Fig. 17). Thus, ADP-ribosylation and phosphorylation in the NTD of AR appear to be independent post-translational modifications.

**ADP-ribose-modified AR is read by the MDs in Parp9.** The Parp7-dependence of AR ADP-ribosylation and Dtx3L/Parp9 binding to AR led us to examine how ADP-ribosylation controls AR-Dtx3L/Parp9 complex assembly. Since Parp9 contains two MDs, it was plausible that ADP-ribosylated Cys residues in AR provide docking sites for the Dtx3L/Parp9 heterodimer. We prepared recombinant Parp9 MDs, both individually, and tandemly arranged, as occurs in the native protein, and tested whether these are sufficient for binding AR from R1881-treated cells. The tandem MDs showed binding to AR that was nearly linear with respect to the amount of cell extract added (Fig. 4a and Supplementary Fig. 18a). By contrast, the individual MDs displayed near background levels of AR binding (Fig. 4a and Supplementary Fig. 18a). Thus, the tandem arrangement of MDs in Parp9 is important for efficient binding to ADP-ribosylated AR.

The Parp9 MDs are most closely related to the MacroH2A-like branch of proteins[22]. This group includes MD2 from Parp14, which by modeling is highly similar to MD1 and MD2 of Parp9 (Supplementary Fig. 18b). Because a Gly-to-Glu change in the Parp14 MD2 abrogates ADP-ribose binding[37] (Supplementary Fig. 18b), we generated comparable reduced function substitutions in Parp9 MD1 (G112E) and MD2 (G311E)[20] to test whether the Dtx3L/Parp9 heterodimer uses MDs to bind ADP-ribosylated AR. Parp9 was engineered for biotinylation with an Avi-tag and co-expressed with GFP-BirA, which enabled selective recovery and detection of WT and mutant Parp9 (Supplementary Fig. 18c). MD substitutions in Parp9 (G112, 311E) eliminated binding to ADP-ribosylated AR (Fig. 4b). The same result was obtained

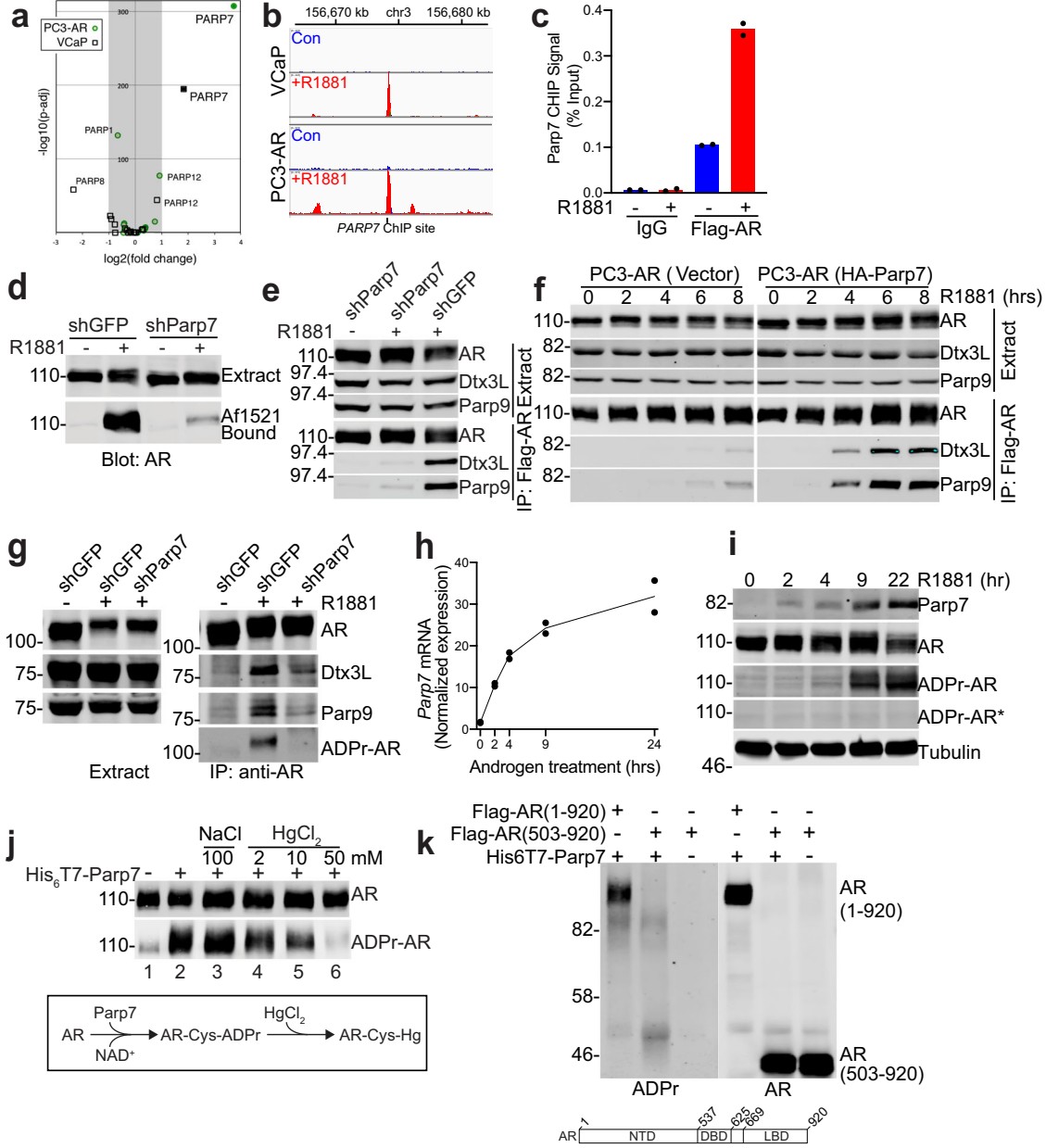

**Fig. 3 Parp7 is the enzyme responsible for AR ADP-ribosylation. a** Volcano plot generated with RNA-seq data[21] from VCaP and PC3-AR showing R1881 effects on Parp expression. The data from all Parps expressed in the two cell lines were plotted, and Parps with adjusted $P$ values $<10^{-30}$ are labeled. Gray shading indicates Parps with log2-fold changes less than $+/-1$. For each treatment group, $n = 3$ biologically independent samples. **b** AR ChIP-seq profiles from VCaP[31] and PC3-AR[32] cells showing androgen-induced AR binding sites in *PARP7*. The basal level of AR binding (R1881-independent) is shown in blue, and the R1881-induced level of AR binding is shown in red. **c** ChIP-PCR of R1881-induced AR binding to the *PARP7* promoter in PC3-AR, using the same color scheme as **b**. **d** Depletion of Parp7 in PC3-AR cells reduces R1881-induced AR ADP-ribosylation. Extracts from PC3-AR cells (stable shGFP and shParp7 cell lines) assayed by Af1521 pulldown and immunoblotting. **e** Depletion of Parp7 in PC3-AR cells reduces AR-Dtx3L/Parp9 complex formation. AR IP'd from shGFP and shParp7 cell line extracts were immunoblotted for AR, Dtx3L, and Parp9. **f** Ectopic expression of Parp7 promotes AR-Dtx3L/Parp9 complex formation. PC3-AR cells stably expressing HA-Parp7, and control PC3-AR cells, were treated with R1881 for 0–8 h, and extracts used for AR IP and immunoblotting. **g** Parp7 mediates AR ADP-ribosylation and complex formation in VCaP cells. VCaP cells (shGFP and shParp7, stable lines) were treated with R1881 overnight. AR was IP'd from cell extracts and probed for AR, Dtx3L, Parp9, and ADP-ribosylation. **h** Androgen induces endogenous Parp7 expression. PC3-AR cells were treated with R1881 (0–22 h) and Parp7 mRNA was measured by RT PCR. **i** Androgen and time-dependent induction of Parp7 expression and AR ADP-ribosylation. PC3-AR cell extracts from an R1881 time course were examined by immunoblotting for Parp7, AR, tubulin, and ADPr in the absence or presence(*) of 10 μM ADPr. **j** ADP-ribosylation of AR with recombinant Parp7 is reversible with HgCl₂. His-tagged Parp7 expressed in insect cells was combined with purified AR, ADP-ribosylated in vitro, treated with NaCl (100 mM) or HgCl₂ (2, 10, 50 mM), and the level of AR ADP-ribosylation detected by Af1521. **k** ADP-ribosylation of AR with recombinant Parp7 requires the N-terminal domain (NTD). Purified His-tagged Parp7 was combined with purified full-length AR (1-920) and AR(503-920) lacking NTD, ADP-ribosylated in vitro, and AR ADP-ribosylation was detected by Af1521. Source data are provided as a Source data file.

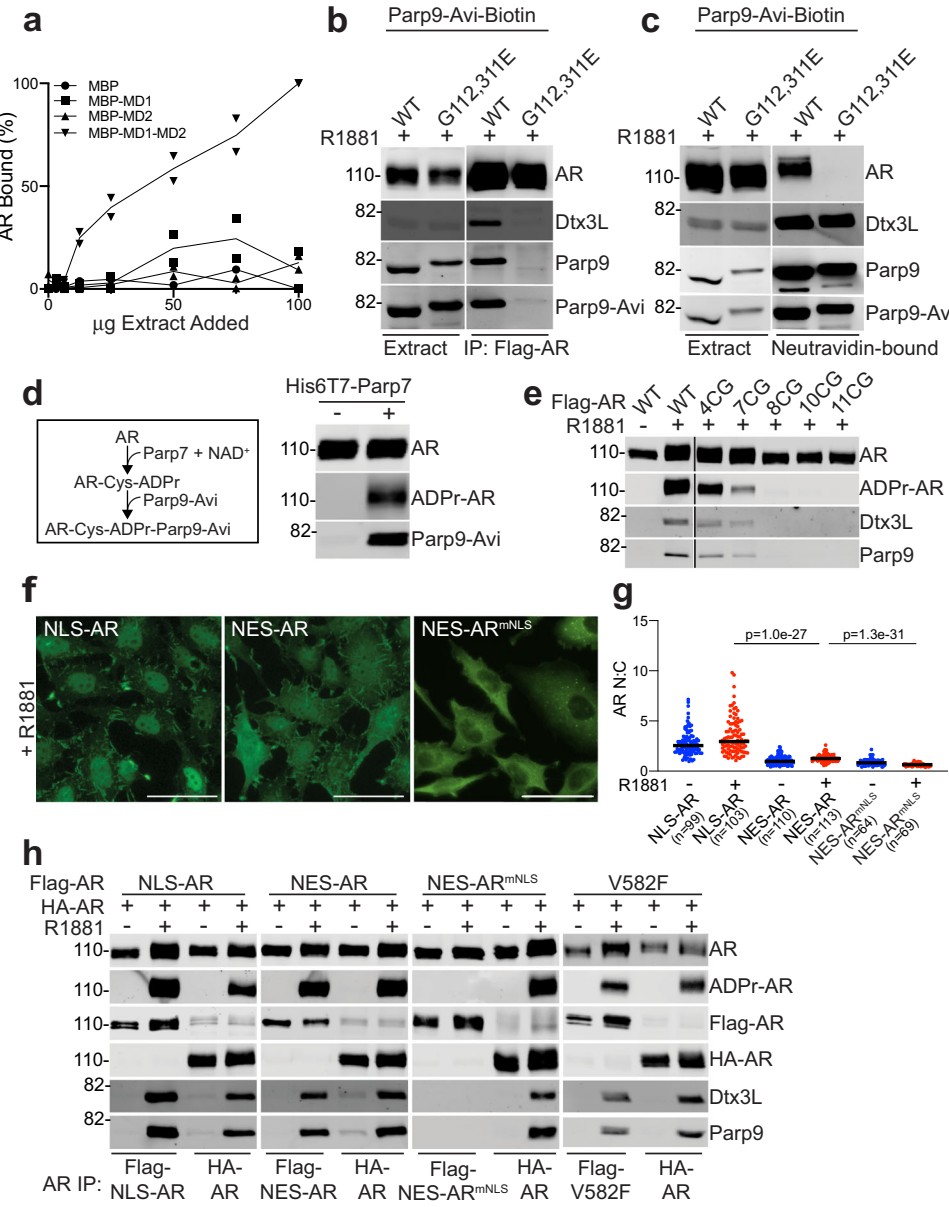

**Fig. 4 Parp9 macrodomains are readers of AR cysteine ADP-ribosylation. a** AR binding to Parp9 MDs immobilized on maltose beads as a function of cell extract concentration (from R1881-treated PC3-AR cells; Supplementary Fig. 18a). Shown are the levels of AR binding to MBP (filled circle), MBP-MD1 (filled square), MBP-MD2 (filled triangle), and MBP-MD1-MD2 (inverted filled triangle). **b** Point mutations in the Parp9 MDs eliminate Dtx3L/Parp9 binding to ADP-ribosylated AR. Stable PC3-AR lines expressing GFP-BirA and C-terminal Avi-tagged WT Parp9 and mutant Parp9 (G112, 311E) were treated with R1881 and used to IP AR. Extracts and bound fractions were examined by immunoblotting for AR, Dtx3L, Parp9, and Parp9-Avi (scheme in Supplementary Fig. 18c). **c** Extracts prepared as described in panel **b**, and Neutravidin beads were used to recover WT and mutant (G112,311E) Parp9-Avi-biotin and associated proteins. The extracts and bound fractions were probed as in panel **b**. **d** ADP-ribosylation of AR by Parp7 in vitro generates Parp9 binding sites. AR immobilized on beads was incubated with $NAD^+ +$ His-tagged Parp7. The beads were washed, combined with extract from Parp9-Avi expressing cells, washed, and the post-reaction bound fractions were probed for ADP-ribosylation and Parp9-Avi. **e** Biochemical analysis of AR ADP-ribosylation mutants. Flag-tagged AR Cys-to-Gly (CG) AR mutants (Supplementary Fig. 19a) were stably expressed in PC3M/HA-Parp7 cells and treated +R1881. AR IP's were probed for AR, AR ADP-ribosylation, Dtx3L, and Parp9. **f** Localization of Flag-tagged AR transport signal fusions in PC3M cells by IF microscopy using M2 antibody. Scale bars = 20 microns. **g** Nuclear:cytoplasmic (N:C) ratios of Flag-tagged AR transport signal fusions in PC3M cells. N:C ratios (−R1881, blue; +R1881, red; median, black line) were measured[39] and analyzed using an unpaired, two-tailed $t$ test. **h** Biochemical analysis of AR transport signal fusions. PC3M cells co-expressing HA-WT AR and Flag-tagged AR with transport signal fusions were prepared as stable lines. HA-WT AR was also co-expressed stably with the DNA-binding domain mutant Flag-tagged AR V582F in PC3M cells (diagrammed in Supplementary Fig. 19b). Cells were treated +R1881 and the extracts were used to IP WT and mutant forms of AR. The IP's were then probed for AR, ADP-ribosylation, Dtx3L, and Parp9. Source data are provided as a Source data file.

when the Dtx3L/Parp9 complex was isolated via the biotinylated Avi-tag and probed for AR (Fig. 4c); this approach confirmed that loss of MD function does not affect Parp9 heterodimerization with Dtx3L. Combining recombinant Parp7, NAD+, and bead-immobilized AR reconstituted ADP-ribosylation and Dtx3L/Parp9 binding from cell extract (Fig. 4d). Collectively, the data indicate that Parp9 MDs function as ADP-ribose readers in the androgen-regulated pathway for AR-Dtx3L/Parp9 complex assembly.

**Parp9 binding is selective for ADP-ribosyl-cysteine.** Eleven ADP-ribosylation sites were identified in AR by MS/MS (Supplementary Table 1 and Supplementary Figs. 1–13). To determine whether one or several ADP-ribosylation sites are linked to androgen-induced assembly of the AR-Dtx3L/Parp9 complex, we generated a panel of cell lines expressing multi-site, Cys-to-Gly substitutions in Flag-AR (Supplementary Fig. 19). In these experiments, HA-tagged Parp7 was co-expressed with the WT and mutant AR proteins to ensure that Parp7 is not limiting. Mutation of all eleven ADP-ribosylation sites (11CG) abolished AR ADP-ribosylation and binding to Dtx3L/Parp9 assessed by co-IP, without affecting R1881-induced nuclear import (Fig. 4e and Supplementary Fig. 20a–c). Comparing the 8CG and 7CG mutants suggested that a low level of ADP-ribosylation and complex formation (~10% of WT) involves the DBD. The 4CG mutant (C125, 290, 327, 406G) showed a reduced level of ADP-ribosylation and complex formation (~50% of WT; Fig. 4e and Supplementary Fig. 20c). This result suggests that Cys residues within the NTD are important acceptor sites for ADP-ribosylation that mediate complex formation. Mutation of pairs of Cys-ADP-ribosylation sites (C125, 290G; C327, 406G) sites did not significantly reduce AR ADP-ribosylation or Dtx3L/Parp9 binding (Supplementary Fig. 20d). This suggests there is site redundancy for AR ADP-ribosylation and complex formation.

**AR ADP-ribosylation is a nuclear event.** Androgen induces AR import into the nucleus 15–30 min after treatment[38]. The time-scale of import compared to ADP-ribosylation (Fig. 3i) suggests the latter is a nuclear event. We tested this by generating AR fusions with transport signals that can drive protein localization to the nucleus (SV40 NLS) and cytoplasm (c-Abl NES; Supplementary Fig. 19b). Using IF microscopy to measure nuclear: cytoplasmic (N:C) ratios[39], we confirmed that the transport signal fusions drive AR localization to the nucleus (NLS-AR) and cytoplasm (NES-AR), though excluding AR from the nucleus also required mutating its endogenous NLS (Fig. 4f, g). Restricting AR to the cytoplasm (NES-AR^mNLS) prevented ADP-ribosylation and, as expected, eliminated complex formation (Fig. 4h). By contrast, constitutively nuclear AR (NLS-AR) and partially nuclear AR (NES-AR) displayed R1881-induced ADP-ribosylation and complex formation that was comparable to WT AR which was co-expressed HA-AR (Fig. 4h). Our data show that nuclear localization of AR is necessary for ADP-ribosylation by Parp7, and that androgen regulation of ADP-ribosylation is separable from androgen regulation of AR import. AR ADP-ribosylation is not strictly dependent on AR association with chromatin since a DNA-binding domain mutant of AR (V582F) can be ADP-ribosylated and undergo complex formation (Fig. 4h).

**Olaparib inhibits AR ADP-ribosylation.** The Parp1/2 inhibitor Olaparib, used for ovarian, breast, and prostate cancer[40], has activity towards Parp7 in vitro[33]. We tested whether Olaparib inhibits Parp7 in cells and potentially affects AR ADP-ribosylation and AR-Dtx3L/Parp9 complex assembly. We found

that androgen-induced AR-Dtx3L/Parp9 complex assembly was inhibited markedly by 0.4 μM Olaparib (Fig. 5a). Olaparib reduced androgen-induced AR ADP-ribosylation detected on blots, and when assayed by Af1521 pulldown (Fig. 5b, c). Olaparib treatment also reduced Parp1 recovery on Af1521 beads, showing it inhibits the basal activity of Parp1 (Fig. 5b). To examine whether Parp1 has a contribution to AR ADP-ribosylation, we co-treated cells with R1881 and Veliparib, another Parp1/2 inhibitor. AR ADP-ribosylation was insensitive to Veliparib at drug concentrations (33–330 nM) that inhibit $H_2O_2$-induced Parp1 automodification (Fig. 5d and Supplementary Fig. 21), suggesting the AR pathway operates independently of Parp1.

Olaparib might act directly on Parp7, given it reduces Parp7 auto-ADP-ribosylation in R1881-treated cells measured by Af1521 pulldown (Fig. 5e). We explored this question further using purified, recombinant Parp7 and measuring auto-ADP-ribosylation in the presence of $^{32}P$-NAD+. We found that Olaparib inhibited Parp7 at a concentration ($K_i = 1.1$ μM; Fig. 5f) comparable to that reported by the Mathews group[33]. We also used Olaparib to query the cellular stability of ADP-ribose conjugated AR. After treating PC3-AR cells with R1881 to induce Parp7-mediated AR ADP-ribosylation, the cells were co-treated with Olaparib to prevent further ADP-ribosylation and CHX to inhibit new protein synthesis. When the level of AR ADP-ribosylation was normalized to AR, we noted a decrease in AR ADP-ribosylation during the first hour that was not accompanied by a reduction in AR protein. The initial decrease was followed by a gradual decline in AR ADP-ribosylation over 6 h (Fig. 5g). We speculate that AR contains both fast and slow turnover ADP-ribosylation sites.

The apparent off-target effects of Olaparib on Parp7 may warrant consideration given that the Parp7 gene is positively associated with overall survival (OS) in ovarian cancer[41]. From prostate cancer RNA-seq data[42,43], Parp7 expression is lower in prostate cancer metastases versus primary prostate tumors, and higher levels of Parp7 are associated with longer disease-free survival (Fig. 5h, i).

**Parp7 and Dtx3L/Parp9 modulate AR-dependent transcription.** The Dtx3L/Parp9 heterodimer can affect transcription, positively or negatively depending on the context. To test whether Dtx3L/Parp9 contributes to AR-regulated gene expression in prostate cancer cells, we engineered VCaP cells with a Dox-inducible shRNA to Dtx3L (Supplementary Fig. 22a) and used the lines for transcriptomic analysis. Control and Dtx3L knockdown (+Dox) VCaP cells were incubated with R1881 for 24 h prior to RNA isolation or left untreated, and RNA-seq was performed on triplicate samples. Principal component analysis (PCA) indicated the four groups clustered separately with the majority of the variance accounted for by PC1, which separated the R1881-treated from the untreated samples (Supplementary Fig. 23a and Supplementary Data 1). Differentially expressed genes were identified by pairwise comparisons using a 0.5 log2-fold change and an adjusted $P$ value cutoff of <0.001. This identified >7000 genes differentially expressed in response to R1881, and 683 genes with different expression levels between control and Dtx3L knockdown, the latter without R1881 treatment (Fig. 6a). Strikingly, >60% of the genes affected by Dtx3L knockdown (439 genes) were also regulated by R1881 (Fig. 6a).

The overlap of R1881- and Dtx3L-regulated genes suggested a significant enrichment for genes that were R1881-activated and had a higher basal expression with Dtx3L knockdown (114 genes), as well as genes that were repressed by both R1881 treatment and Dtx3L knockdown (208 genes) (Fig. 6b, left).

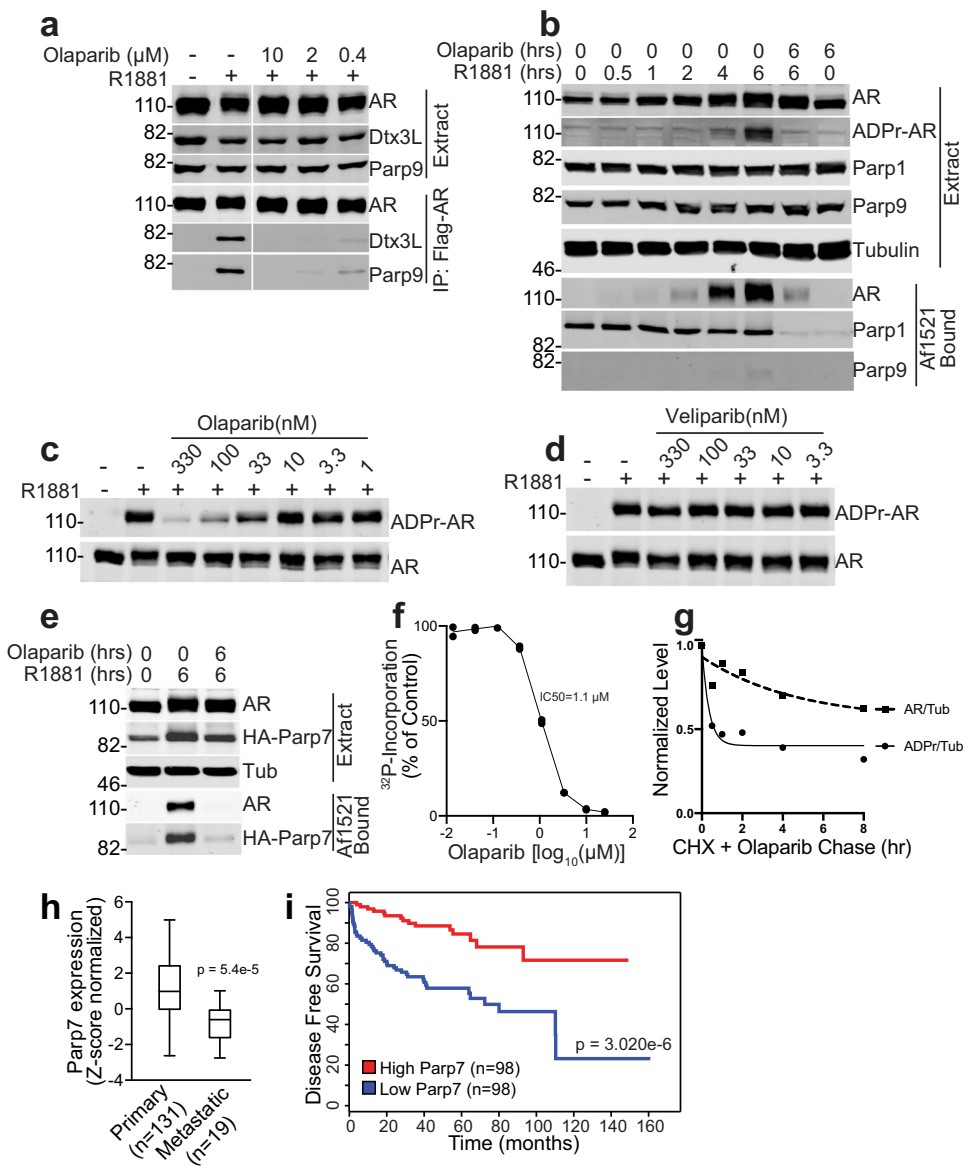

**Fig. 5 Olaparib inhibits AR ADP-ribosylation and complex assembly. a** Olaparib inhibits androgen-induced AR-Dtx3L/Parp9 complex formation. PC3-AR cells were treated with R1881 (2 nM) and the concentrations of Olaparib indicated. Extracts were used to IP AR, and the bound fractions were probed for AR, Dtx3L, and Parp9. **b** Olaparib inhibits androgen-induced AR ADP-ribosylation. PC3-AR cells were treated with R1881 and Olaparib in the combinations and times indicated. Cell extracts were used for Af1521 pulldowns, and the extracts and bound fractions were analyzed by blotting for AR, ADPr-AR, Parp1, and Parp9. **c** Concentration dependence of Olaparib inhibition of AR ADP-ribosylation. PC3-AR cells were co-treated with R1881 and the indicated concentrations of Olaparib, and the levels of AR and ADP-ribosylation examined by blotting. Olaparib inhibition of Parp1 automodification was verified by immunoblotting (Supplementary Fig. 21a). **d** The Parp1/2 inhibitor Veliparib does not inhibit AR ADP-ribosylation. PC3-AR cells were co-treated with R1881 and the indicated concentrations of Veliparib, and the levels of AR and ADP-ribosylation examined by blotting. Veliparib inhibition of Parp1 automodification was verified by immunoblotting (Supplementary Fig. 21b). **e** Parp7 automodification in cells and binding to Af1521 beads is inhibited by Olaparib. PC3-AR-HA-Parp7 cells were co-treated with R1881 and Olaparib for the indicated times, and cell extracts were analyzed by Af1521 pulldown and immunoblotting for AR and HA-Parp7. **f** Parp7 automodification in vitro is inhibited by Olaparib. Recombinant Parp7 expressed and purified from insect cells was incubated with $^{32}$P-NAD$^+$ with a range of Olaparib concentrations, the level of automodification quantified and plotted after autoradiography. **g** Olaparib chase experiment indicates that AR has ADP-ribosylation sites with fast and slow turnover. PC3-AR cells were treated with R1881 overnight, and subsequently chased with Olaparib (10 μM) and CHX (100 μg/ml) to prevent further ADP-ribosylation and AR synthesis, respectively. The levels of AR and AR ADP-ribosylation were measured and plotted as a function of tubulin level. **h** Parp7 expression levels (Z-score normalized, RNA-seq data) in primary and metastatic prostate tumors. The box shows the median and the upper and lower quartiles, and the whiskers indicate the minimum and maximum. Expression levels were compared by unpaired, two-tailed *t* test. Further details of the analysis are contained in "Methods". **i** Prostate cancer patient outcome stratified by Parp7 expression levels (RNA-seq). Plotted is the time of disease-free survival for patients with high Parp7 levels (red) versus low Parp7 levels (blue). *P* value determination by a log-rank test is shown. Source data are provided as a Source data file.

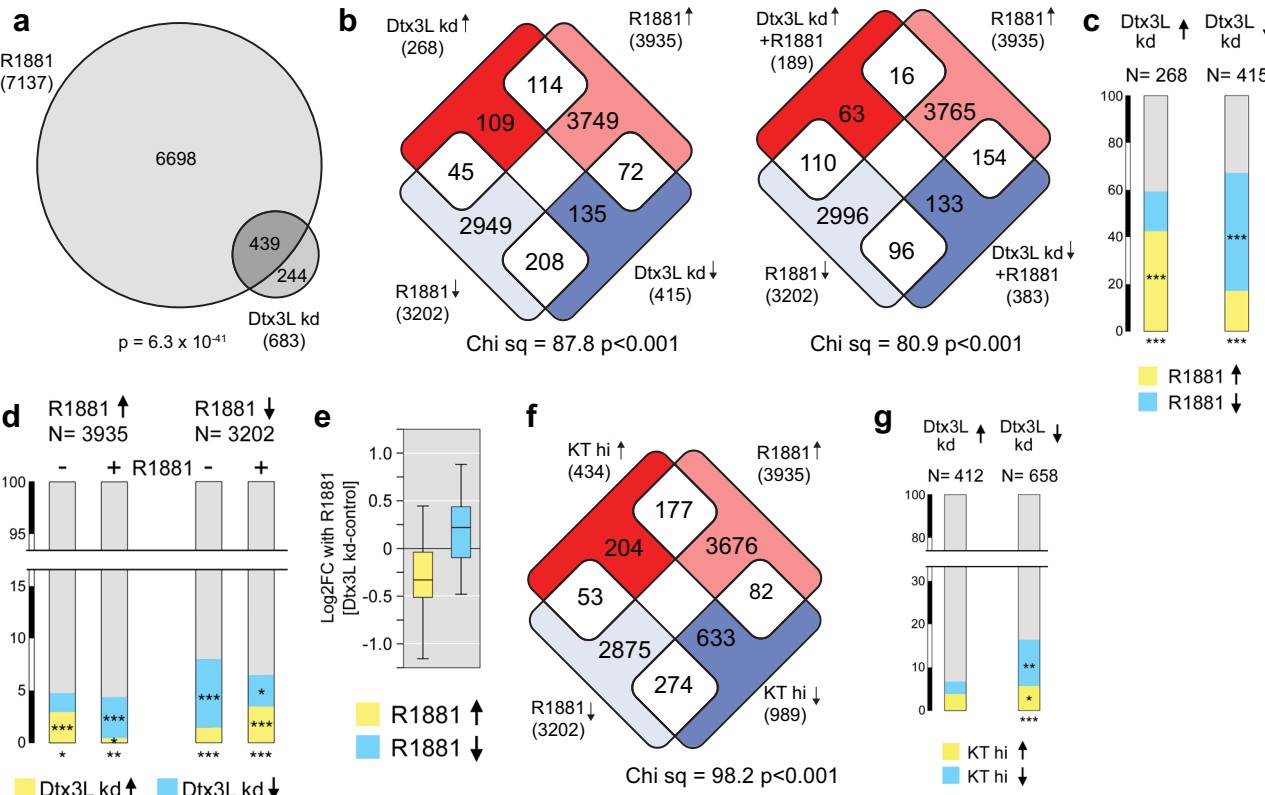

**Fig. 6 Dtx3L levels affect AR-regulated transcription. a** Overlap between genes with altered expression (RNA-seq) after 24 h of R1881 treatment and by Dtx3L knockdown in VCaP cells. The *P* value for the overlap is from a hypergeometric test (1.63-fold overrepresentation). For each treatment group, $n = 3$ biologically independent samples. **b** Genes increased and decreased (up, down arrows) in VCaP cells after R1881 treatment and Dtx3L knockdown intersected to identify genes that change significantly in both sets, comparing the Dtx3L effect −R1881 (left) and +R1881 (right). *P* values are from the Chi-squared test using a 2 × 2 contingency table for the distribution of overlapping genes. **c** Genes (%) activated or repressed by Dtx3L knockdown in VCaP cells that are activated or repressed by R1881. ***$P < 0.001$ determined by Chi-squared test of a 2 × 2 contingency table for comparison to the expected overlap for activated or repressed genes or all R1881-regulated genes. **d** For genes that are activated or repressed by R1881 the percentages that are affected by Dtx3L knockdown +R1881 in VCaP cells are shown. *$P < 0.05$, **$P < 0.01$, ***$P < 0.001$ for comparison to the expected overlap for activated or repressed genes or all Dtx3L-regulated genes. *P* values were determined by the Chi-squared test of a 2 × 2 contingency table. **e** The effect of Dtx3L knockdown on R1881-activated and repressed genes in VCaP cells is shown as the differential effect of R1881 between control and Dtx3L knockdown. **f** Genes with increased or decreased expression +R1881 (VCaP cells) were intersected with those genes with higher or lower expression (up, down arrows) in the top versus bottom decile (KT hi) of the TCGA PrAd data sorted by combined *KLK3* and *TMPRSS2* expression. *P* values as in panel **b**. **g** The percentage of Dtx3L-regulated genes differentially expressed (VCaP cells) in the top versus bottom decile (TCGA PrAd data) sorted by combined *KLK3* and *TMPRSS2* expression. *P* values (*$P < 0.05$, **$P < 0.01$, ***$P < 0.001$) were determined by Chi-squared test of a 2 × 2 contingency table. Source data are provided as a Source data file.

R1881-regulated genes were significantly enriched among those with a higher expression upon Dtx3L knockdown, driven by the greater than expected proportion of R1881-activated genes (Fig. 6c). Conversely, the overrepresentation of R1881-regulated genes among those with lower expression in the setting of Dtx3L knockdown was primarily due to R1881-repressed genes. Comparing the effect of Dtx3L knockdown in the presence of R1881 indicated a significant bias toward a higher expression of R1881-repressed genes (110 genes) and lower expression of R1881-induced genes (154 genes) (Fig. 6b, right). Thus, Dtx3L has different effects on the basal and activated expression levels of R1881-regulated genes, and this is different for genes that are normally activated and repressed by androgen. Although the proportion of R1881-regulated genes affected by Dtx3L knockdown is <10%, this is more than expected by chance, and the distribution of Dtx3L knockdown-dependent effects suggests that Dtx3L depletion primarily increases the basal activity of R1881-activated genes and reduces expression in the presence of R1881 (Fig. 6d). By contrast, for R1881-repressed genes, Dtx3L knockdown generally lowered the basal expression and increased

expression in the presence of R1881. This is consistent with Dtx3L promoting R1881-responsive transcriptional effects, such that knockdown blunts the R1881 effect on both activated and repressed genes. In support of this view, the fold induction by R1881 of genes that are activated by R1881 and sensitive to Dtx3L knockdown is lower in the Dtx3L knockdown cells, and fold repression of R1881-repressed genes is also reduced by a small but significant amount (Fig. 6e and Supplementary Fig. 23b). Thus, Dtx3L knockdown has opposing effects on R1881-activated and repressed genes, effectively reducing the fold change in each group. This is consistent with a Dtx3L function that involves increasing the magnitude of the androgen response for a subset of AR-related genes.

To compare the effects of Dtx3L knockdown in VCaP cells to human prostate cancer, we ranked TCGA prostate cancer gene expression data using two well-characterized AR-regulated genes, *KLK3* and *TMPRSS2*. We then compared gene expression between the upper and lower deciles based on *KLK3* and *TMPRSS2*, and overlapped the genes with higher or lower expression ($P < 10^{-7}$, log2-fold difference +1) with the data from

R1881-treated VCaP cells. This revealed an enrichment in the overlap between R1881-induced and higher expression in the *KLK3* and *TMPRSS2* upper 10% (Fig. 6f), which indicates this is a suitable proxy for androgen regulation in patients samples. This was supported by GSEA, which showed enrichment for the Androgen response in the top 10% compared to the bottom 10% of samples ranked by *KLK3* and *TMPRSS2* (Supplementary Fig. 23c). Overlapping the genes that were differentially expressed between the TCGA top and bottom deciles by *KLK3* and *TMPRSS2* expression with genes affected by Dtx3L knockdown showed enrichment for androgen-regulated genes from the TCGA data among those that were downregulated by Dtx3L knockdown (Fig. 6g and Supplementary Table 2). This analysis further supports the finding that Dtx3L knockdown preferentially affects androgen-regulated gene expression.

We then examined the contribution of Parp7 to genes regulated by Dtx3L and androgen. Comparing patient data stratified by *KLK3* and *TMPRSS2* and VCaP data from the Dtx3L knockdown, we selected four genes for the query. Using stable VCaP lines (shGFP and shParp7; Supplementary Fig. 22c), we found the androgen-induced levels of *KLK3*, *MYBPC1*, and *MESP1* were significantly reduced by Parp7 depletion (Fig. 7a).

The *KLK3* (*PSA*) promoter is used extensively by the prostate cancer field to measure AR-dependent transcription. We employed the PSA-luciferase reporter, which contains proximal and distal androgen response elements (AREs) to test whether AR-Dtx3L/Parp9 complex formation regulates transcription (Fig. 7b). We transfected the PSA reporter into PC3-AR-Parp9 knockout cells, along with WT Parp9 and mutant Parp9 bearing point mutations in the MDs (G112, 311E; Supplementary Fig. 18b, c) that eliminate Dtx3L/Parp9 binding to ADP-ribosylated AR (Fig. 4b, c). We found that R1881 induction of AR-dependent transcription was enhanced by reconstitution with WT Parp9, but not by the MD mutant (G112, 311E) of Parp9 (Fig. 7c). This result complements the RNA-seq data showing that Dtx3L knockdown selectively affects a subset of R1881-regulated genes, and it provides evidence that the MD function of Parp9 is important for enhancing AR-dependent transcription. Similarly, mutation of four AR ADP-ribosylation sites (4CG mutants) reduced androgen-induced transcription, as did the 7CG and 8CG mutants (Fig. 7d). These data link Parp9 recognition of Cys-ADP-ribosylation to AR activity.

**ADP-ribosylated AR is recruited to chromatin**. A model suggested by our data is that androgen signaling induces Parp7 expression, Parp7 directly modifies AR with ADP-ribose, ADP-ribosylated AR is recognized by Dtx3L/Parp9 via MDs, and the AR-Dtx3L/Parp9 complex modulates a subset of AR target genes. We used the *MYBPC1* gene to further explore this model. *MYBPC1* is part of a gene signature that can distinguish Gleason score 7 from the more aggressive Gleason score >8 prostate cancers as part of the Decipher classifier[44]. *MYBPC1* is an AR target gene and shows robust binding in response to androgen in VCaP and PC3-AR cells[31,32] (Fig. 8a), and androgen induction of *MYBPC1* mRNA in VCaP cells was reduced by Parp7 depletion (Fig. 7a and Supplementary 22c). Depletion of Parp7 also reduced R1881-induced transcription of *MYBPC1* during a time course of androgen treatment of PC3-AR cells (Fig. 8b, c and Supplementary 22b). To determine whether AR bound to the *MYBPC1* promoter is ADP-ribosylated, we used Af1521 for a ChIP-type approach to detect chromatin-associated ADP-ribosylation (ADPr ChAP)[45]. R1881-induced ADP-ribosylation was detected at two AR binding sites in the *MYBPC1* gene, but not at an intergenic site (Fig. 8d). To establish whether Af1521 is detecting ADP-ribosylated AR bound to the MYBPC1 promoter, we

performed parallel AR ChIP and ADPr ChAP, and sequential ADPr ChAP-AR ChIP (Fig. 8e). By AR ChIP, AR undergoes R1881-induced binding to site 2 in *MYBPC1*. AR binding to site 2 was not inhibited by treating cells with Olaparib (Fig. 8f). Parallel ADPr ChAP showed that site 2 enrichment on Af1521 was higher using chromatin from R1881-treated cells, it was abolished if cells were treated with Olaparib, and it was lost if ADP-ribose was included during the Af1521 binding step (Fig. 8g). For sequential ADPr ChAP-AR ChIP, chromatin bound to Af1521 was eluted with ADP-ribose and used for a "re-chip" step with Flag antibody in the absence and presence of excess Flag peptide. Because this enabled specific recovery of Flag-tagged AR bound to chromatin (Fig. 8g, right panel), the sequential ADPr ChAP-AR ChIP is consistent with chromatin binding of ADP-ribosylated AR. Inhibition of AR ADP-ribosylation with Olaparib reduces R1881-induced *MYBPC1* expression ($P < 0.001$). By contrast, treating cells with Veliparib, which inhibits Parp1 but not Parp7, does not significantly change AR ADP-ribosylation nor *MYBPC1* gene expression (Fig. 8h). These data establish that ADP-ribosylated AR can be recruited to ARE sites, but ADP-ribosylation is not essential for chromatin binding.

## Discussion

Our studies have revealed a regulatory mechanism for androgen signaling based on post-translational modification of AR by ADP-ribosylation. The pathway is initiated by androgen induction of the ADP-ribosyltransferase *Parp7*, a direct AR target gene. Parp7 conjugates ADP-ribose to AR on a total of eleven sites, seven of which map to the NTD. ADP-ribose moieties on AR are subsequently recognized by tandem MDs in Parp9, which forms a stable heterodimer with the ubiquitin E3 ligase Dtx3L. ADP-ribosylation of AR results in highly selective E3 recruitment and modulation of a subset of androgen-regulated genes in prostate cancer cells. Thus, Parp7 acts as an androgen-inducible writer of ADP-ribose, and Parp9 provides the reader function.

These data show that Parp7 modification of AR is restricted to Cys residues and that the reaction occurs in the nucleus. The site assignment is based on MS/MS and on biochemical data showing that the ADP-ribosylation sites in AR are sensitive to mercury, but not to reagents that release ADP-ribose from other amino acids. While we cannot rule out the possibility that Parp7 labels other amino acids in AR, it should be noted that all AR ADP-ribosylation detected with Af1521 was eliminated by mutating the 11 Cys sites identified by MS/MS. The apparent selectivity of Parp7 for Cys residues in AR is interesting in light of the fact that Parp7 itself is ADP-ribosylated on Cys residues[33], including two additional sites reported in this study. In the AR, six of the ADP-ribosylation sites map to the AF1 region, which is known to be critical for the transcription function of AR.

MDs are phylogenetically conserved protein modules that bind mono- and poly-ADP-ribose, and in some cases, show hydrolysis activity[22]. Parp family members Parp9, Parp14, and Parp15 contain two or more copies of the macroH2A-type MD within their respective N-terminal regions. We determined that the tandem MDs in Parp9 recognized the ADP-ribose that is conjugated to Cys residues in AR. Our data suggest that both MDs in Parp9 are necessary for efficient recognition of ADP-ribosylated AR. Our data are reminiscent of other tandemly arranged protein modules, such as SUMO interaction motifs, that use multivalent contact to drive highly selective protein–protein interactions[46]. The presence of seven acceptor sites for Cys-ADP-ribosylation in the AR NTD implies redundancy that ensures the Dtx3L/Parp9-AR interaction, or it might reflect alternative site utilization by Parp7, which is known to occur with other enzyme pathways. The MDs of Parp9 probably have reader function in the context of

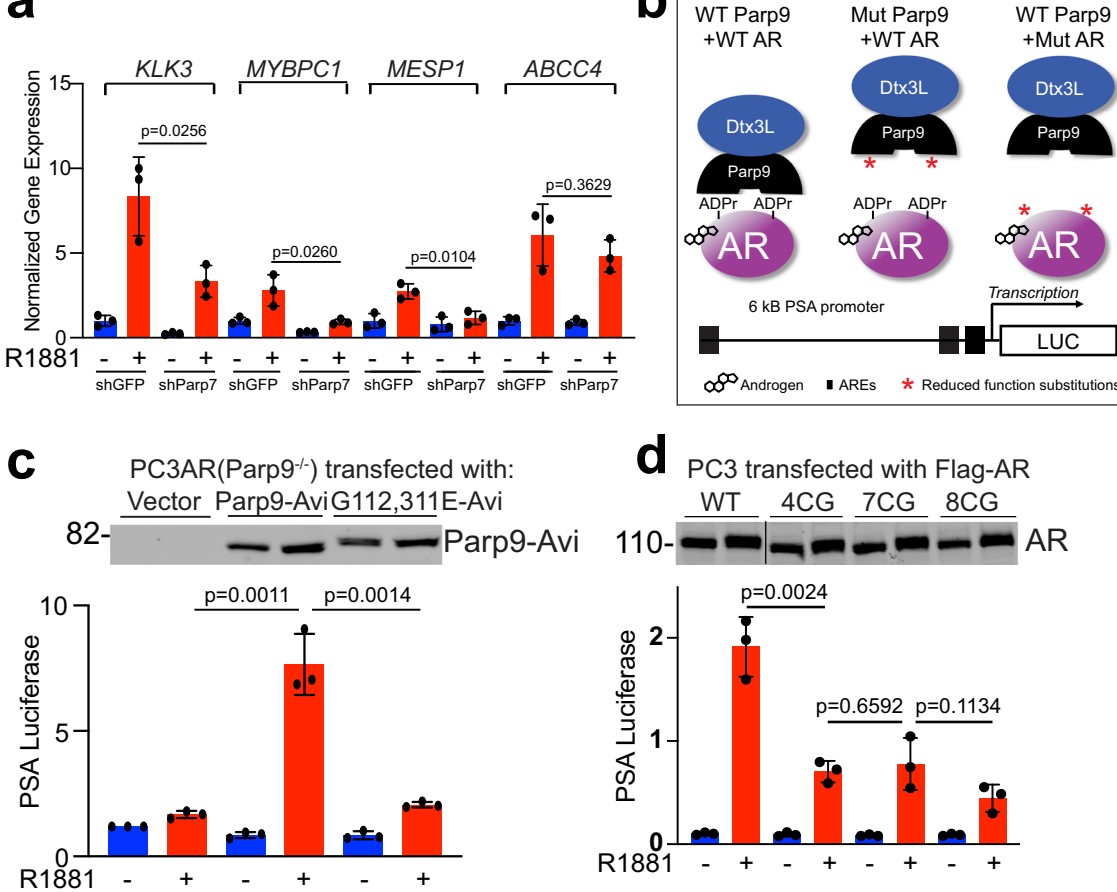

**Fig. 7 ADP-ribosylation and macrodomain function contribute to AR-dependent transcription. a** qPCR of AR target genes in control (shGFP) and Parp7-depleted (shParp7) VCaP cells, untreated (blue) and androgen-treated (red). The data were analyzed by an unpaired, two-tailed *t* test and are plotted as the mean + s.d. Dots represent individual values. For each treatment group, $n = 3$ biologically independent samples. **b** Scheme for testing the effects of Parp9 MD and AR ADP-ribosylation mutants on androgen-induced transcription from the PSA promoter. **c** Parp9 MD function is required for AR-dependent transcription. Parp9 knockout cells (PC3-AR background) were reconstituted with WT and mutant (G112, 311E) forms of Parp9, and AR activity measured using the 6-kB *PSA* promoter fused to firefly luciferase. Data were normalized to CMV-Renilla luciferase and Parp9 levels examined by immunoblotting. Color scheme is the same as **a**. The data were analyzed by an unpaired, two-tailed *t* test and are plotted as the mean + s.d. Dots represent individual values. For each treatment group, $n = 3$ biologically independent samples. **d** Mutation of ADP-ribosylation sites reduces AR-dependent transcription. AR-negative PC3M-HA-Parp7 cells were reconstituted with WT and mutant (4CG, 7CG, 8CG) forms of AR, and analyzed as described in **c**. The data were analyzed by an unpaired, two-tailed *t* test and are plotted as the mean + s.d. The color scheme is the same as **a**. Dots represent individual values. For each treatment group, $n = 3$ biologically independent samples. Source data are provided as a Source data file.

interferon (IFN) signaling in immune cells, since Dtx3L/Parp9-STAT1 complexes isolated by IP can be dissociated by the addition of ADP-ribose[47].

Previous work from our group[20] showed that the Dtx3L/Parp9 heterodimer contains ADP-ribosyltransferase activity, which is in addition to the E3 ubiquitin ligase activity discovered by the Shipp laboratory[19]. Both activities are utilized in the context of Ub processing and conjugation, where the Dtx3L/Parp9 heterodimer can transfer ADP-ribose to the carboxyl terminus of Ub and block substrate conjugation[20]. The outcome of the reaction, substrate conjugation or Ub-ADP-ribosylation, is dictated by $NAD^+$ concentration, with the latter favored in the presence of high $NAD^+$. We found that E3 and ADP-ribosyltransferase activities are functional in the AR-Dtx3L/Parp9 complex, suggesting Parp7 modification of AR can target these activities to chromatin sites. This view is consistent with data showing that Dtx3L/Parp9 depletion affects the expression of certain AR-regulated genes, and ADPr ChAP showing ADP-ribosylated AR is recruited to chromatin. We also employed a PSA reporter assay to test whether the Dtx3L/Parp9 binding to AR promotes

transcription. Using Parp9 knockout cells, we found that androgen induction of AR-dependent transcription from the PSA promoter was significantly increased by WT Parp9, but not by Parp9 that was deficient for binding ADP-ribosylated AR because of point mutations in its two MDs. These data argue that a physical interaction between the Dtx3L/Parp9 heterodimer and AR is important for transcription. Consistent with this view, Cys-to-Gly mutations in four of the ADP-ribose acceptor sites in the AR NTD reduced ADP-ribosylation, Dtx3L/Parp9 complex formation, and AR-dependent transcription from the PSA promoter. Exactly how Dtx3L/Parp9 promotes transcription remains to be determined. One scenario is that it involves the E3 ligase activity of Dtx3L, which is known to modify histones, but an effect on co-factor interactions with AR is also possible. AR binding to chromatin appears to be independent of ADP-ribosylation, based on our finding that Olaparib inhibition of AR ADP-ribosylation did not detectably reduce AR binding to the MYBPC1 promoter.

Dtx3L has been shown to act as an E3 ligase for histone monoubiquitylation and contribute to the regulation of chromatin-based events. Histone modification by Dtx3L, notably

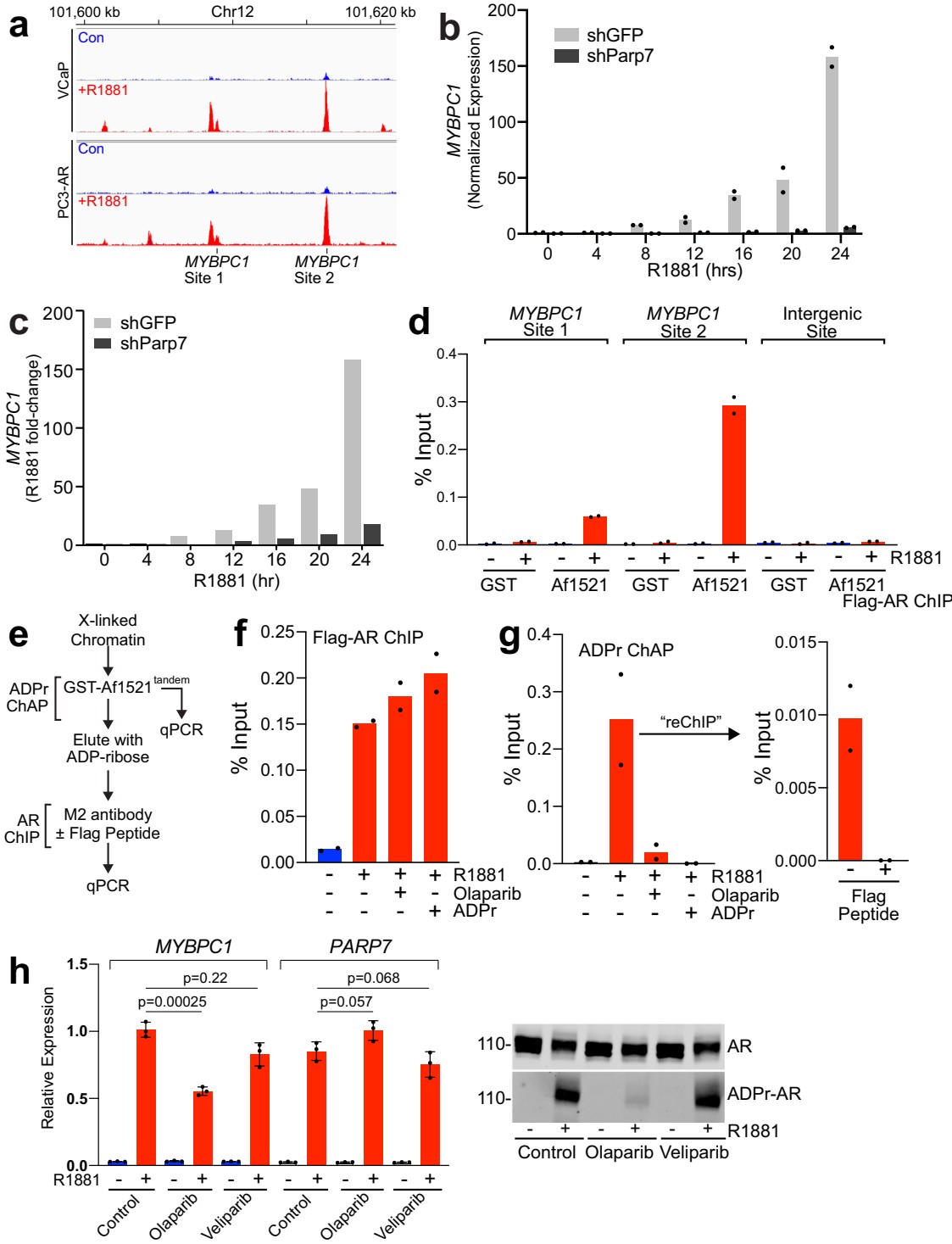

**Fig. 8 ADP-ribosylated AR is recruited to the *MYBPC1* gene promoter. a** AR ChIP-seq profile from VCaP and PC3-AR cells showing AR binding to sites in the *MYBPC1* gene in the control (blue) and R1881-treated (red) cells. **b** *MYBPC1* mRNA induction (qPCR) as a function of androgen treatment time in shGFP (gray) and shParp7 (black) PC3-AR cells. **c** Data from **b** plotted as fold-change relative to the 0-h timepoint. **d** ADPr ChAP using immobilized GST-Af1521-beads and the primer sites indicated. Color scheme is the same as **a**. **e** Scheme for sequential ADPr ChAP and AR ChIP. **f** AR ChIP (Flag antibody, *MYBPC1* site 2) using cells treated with R1881 and Olaparib. Color scheme is the same as **a**. **g** ADPr ChAP and AR ChIP (*MYBPC1* site 2) using R1881-treated cells. Recovery of ADP-ribosylated chromatin (Site 2) on Af1521 beads is virtually eliminated when cells are treated with Olaparib, or when ADPr is added during the ChAP step. Color scheme is the same as **a**. **h** RT PCR of *MYBPC1* and *PARP7* expression (left panel) and detection of AR protein and AR ADP-ribosylation (right panel) in cells treated with Olaparib and Veliparib. Color scheme is the same as **a**. The data was analyzed by an unpaired, two-tailed *t* test and are plotted as the mean + s.d. Dots represent individual values. For each treatment group, *n* = 3 biologically independent samples. Source data are provided as a Source data file.

histone H4K91-Ub generation, contributes to DNA repair kinetics by enhancing 53BP1 recruitment[48]. In another context, Dtx3L/Parp9 mediates STAT-dependent repression of the tumor suppressor IRF1, consistent with the repressive effect of Dtx3L/Parp9 in tethering assays[18,47,49]. By contrast, Dtx3L/Parp9 enhances IFN-dependent gene expression through STAT1 by utilizing its E3 activity towards histone H2BJ[50]. Our gene expression analysis indicates that Dtx3L/Parp9 depletion results in negative and positive effects on AR-dependent transcription, depending on the gene. Dtx3L/Parp9 levels influence the basal levels of AR-regulated genes, assayed in the absence of androgen, as well as the induced levels of androgen-regulated genes, measured in response to R1881. The crosstalk between IFN and androgen signaling appears to be bidirectional given that IFN can induce Dtx3L/Parp9, and AR can repress IFN signaling through induction of genes such as SOCS3 in prostate cancer cells[51]. The AR-Dtx3L/Parp9 can therefore be considered a signaling node that is regulated by at least three inputs: (i) androgen, which increases pathway activity by inducing Parp7 expression; (ii) $NAD^+$ levels, which influence the level of substrate conjugation with Ub; (iii) IFN, which controls Dtx3L and Parp9 expression through a STAT-regulated bidirectional promoter[52]. It seems probable that these diverse inputs help fine-tune transcription of AR-regulated genes. Based on our RNA-seq data, Dtx3L and Parp9 may also exert effects on transcription that are androgen-independent, and AR transcriptional effects that are ADP-ribosylation independent.

RNA-seq data from prostate adenocarcinoma (PrAd) patients suggest there is an inverse correlation between Parp7 expression and the time period between therapy and biochemical recurrence. We also noted in RNA-seq data that Parp7 levels are lower in prostate cancer metastases than in primary tumors. These observations, which suggest a reduction in Parp7 mono-ADP-ribosylation activity could be associated with disease progression, is interesting given the related findings in ovarian cancer. Amplification of the PARP7 gene was associated with an overall survival benefit (44.8 versus 15.7 months) in ovarian cancer patients[41]. Also, a genome-wide association study identified an ovarian cancer susceptibility locus in the PARP7 gene[53]. These genomic data prompt the hypothesis that Parp7 activity could favor clinical outcome, and that Parp inhibitors lacking off-target effects on Parp7 might be more beneficial than pan-Parp inhibitors.

Parp7 makes important contributions to other nuclear pathways. In the liver, Parp7 (termed Tiparp, for 2,3,7,8-tetra-chlorobenzo-p-dioxin-inducible poly-ADP-ribose polymerase) is induced by the aryl hydrocarbon receptor (AHR) and helps protect against dioxin-induced lethality[54]. Parp7 restrains AHR transcription activity through ADP-ribosylation, but the mechanism has not been fully defined. Among the dioxin and AHR-regulated genes restrained by Parp7 are the cytokines Il1b, Cxcl2, and Tnf, which point to a role in the inflammatory response[54]. Parp7 also restrains type I IFN signaling through ADP-ribosylation of the kinase TBK1 and suppression of IFN expression[55]. With regard to nuclear receptors, liver X receptor (LXR) can be ADP-ribosylated by Parp7, and Parp7 knockout mice show reduced ligand induction of the LXR target gene Srebp1[56]. Ligand regulation of Parp7 expression and transcription factor ADP-ribosylation appear to be a feature of several nuclear receptor pathways. These data raise the question of whether the ADP-ribose reader and E3 functions of the Dtx3L/Parp9 heterodimer contributes to transcriptional regulation in a variety of biological settings.

## Methods

**Cell culture**. VCaP prostate cells (ATCC CRL-2876) were maintained in DMEM/F12 supplemented with 10% FBS and 1% Pen Strep. HEK293T cells (ATCC CRL-3216) were maintained in medium DMEM supplemented with 5% FBS and 1% Pen Strep. PC3 (ATCC CRL-1435), PC3M (RRID:CVCL_9555), and derivative cell lines were grown in RPMI 1640 supplemented with 5% FBS and 1% Pen Strep. All cells were grown in the presence of 5% $CO_2$ and at 37 °C. For RNA-seq, qPCR, ChIP, and ChAP experiments, cells were switched to phenol red-free medium and FBS depleted of androgens 24 h before R1881 addition.

**Cell line generation using lentiviruses**. HEK293T cells were grown to ~70% confluency and co-transfected with target plasmid and accessory plasmids pMD2g and psPAX2 (ratio 2:1:1) using Fugene-6. After cell incubation at 37 °C for ~16 h, cells were replenished with fresh DMEM supplemented with 35% FBS, and further incubated for another 24 h. The medium containing lentiviruses was transferred to a fresh conical tube, clarified by centrifugation ~700 × g, and then passed through a sterile 0.45-μm filter, with optional concentrating lentivirus using Takara's Lenti-X Concentrator. Infection of target cells was carried out in the presence of 8 μg/ml of polybrene. After 2–3 cell doublings, the cells were either selected with 0.2–0.5 mg/ml of hygromycin or 1–2 μg/ml of puromycin based on the antibiotic resistance of the plasmid, or cell sorted for GFP co-expression. PC3-AR cells expressing Flag-tagged AR was generated and selected by growth in hygromycin. PC3M-HA-AR and PC3M-HA-Parp7 were produced from PC3M cells with an HA-AR and HA-Parp7 lentivirus containing a puromycin resistance marker. Additional cell lines generated were VCaP(shGFP), VCaP(shParp7), and VCaP(pLKO-Tet-ON shVector), VCaP(pLKO-Tet-ON/shDtx3L). Cell lines containing two transgenes were produced by sequential transfection and drug selection. The cell lines constructed with two transgenes were PC3AR(pWPI), PC3AR(pWPI/HA-Parp7), PC3AR(pWPI/Parp9 wild type or mutants-Avi/GFP-BirA*), PC3AR(shGFP), PC3AR(shParp7), PC3M-HA-AR(Flag-AR wild-type and Cys-to-Gly mutants), and PC3M-HA-Parp7(Flag-AR wild-type and Cys-to-Gly mutants).

**Knockout cells**. To generate PARP9-deficient PC3-AR prostate cancer cells, two single-guide RNAs (Supplementary Table 3) targeting exon 4 of the human PARP9 gene (188-bp spacing) were cloned into pX330 vector expressing a human codon-optimized SpCas9 endonuclease (Addgene #42230). PC3-AR cells were transfected with control (empty pX330) plasmid, or with pX330 with sg-PARP9 plasmids along with a pMSCV vector containing the puromycin resistance gene (Clontech) using Lipofectamine 2000 (Invitrogen). Twenty-four hours following transfection, cells were treated with puromycin (2 μg/ml) for 48 h, and were subsequently seeded at low density to obtain single colonies. Individual PC3-AR clones were selected, propagated in culture, and examined for PARP9 deletion by genotyping. Genotyping was performed by PCR amplification of a genomic fragment flanking the two predicted Cas9 cleavage sites within exon 4 of PARP9 (Supplementary Table 3). PARP9 deletion in the edited clones was further confirmed by immunoblotting with PARP9-specific antibodies.

**Antibodies**. The following antibodies were used at the concentrations and dilutions indicated for Western blot (WB) and immunofluorescence microscopy (IF). A rabbit polyclonal was raised against a Parp7 peptide (amino acids 119–132) and affinity-purified before use (WB: 1 μg/ml). The rabbit antibodies to AR (WB: 1 μg/ml), Dtx3L (WB: 0.2 μg/ml), Parp9 (WB: 1 μg/ml), and AR phosphor-Serine-sites (94, 256, 308, and 424) (WB: 1:1000–2000) have been described[20,57]. Commercial antibodies used were AR phosphor-Serine 81 (WB: 1:1000, Sigma 07-1375), M2 (WB: 0.75 μg/ml, IF: 1:1000, Sigma A2220), Tubulin (WB: 1:10,000–20,000, mouse mAb TUB-1A2, Sigma T9028), AviTag (WB: 1 μg/ml, GenScript A00674-100), Parp1 (WB: 1:1000, rabbit mAb E102, Abcam ab32138), HA tag (WB and IF: 1:1000, mouse mAb 16b12, Covance A488-101L; IF: 1:500, rabbit mAb C29F4, Cell Signaling Technology 3724 S), Alexa Fluor 680 donkey anti-Rabbit IgG(H + L) (WB: 1:20,000, Invitrogen A10043), IRDye800-conjugated anti-mouse IgG(H&L) (WB: 1:20,000, Rockland 610-132-121), Cy™3 AffiniPure Donkey Anti-Rabbit IgG (H + L) (IF: 1:400, Jackson Immuno Research Laboratories, Inc. 711-165-152) and Alexa Fluor® 488 AffiniPure Donkey Anti-Mouse IgG (H + L) (IF: 1:200, Jackson Immuno Research Laboratories, Inc. 715-545-150), and Neutravidin-DyLight-800 (WB: 1:20,000, Pierce 22853). Uncropped scans of blots are available in the Source Data file.

**Plasmids**. A lentiviral plasmid encoding Flag-AR and hygromycin resistance[25] was used to generate a panel of Flag-AR mutants (Flag-F805S, Flag-V582F, Flag-C125/290 G, Flag-C327/406 G, Flag-4CG, Flag-7CG, Flag-8CG, Flag-10CG, Flag-11CG, Flag-NLS$^{SV40}$-AR, Flag-NES$^{C-Abl}$-AR, Flag-NES$^{C-Abl}$-AR$^{NLS-Mut}$. HA-AR was constructed by replacing the Flag-tag and hygromycin resistance markers with HA and puromycin resistance, respectively. Af1521 cDNA clone (#AfCD00370825) was obtained from the DNASU Plasmid Repository and cloned into the pGEX-4T-2 bacterial expression vector. To generate tandem Af1521 expression, Af1521 was amplified by PCR and inserted in-frame into the AccI and NotI restriction sites[25]. Other plasmids used were pWPI (Addgene 12254), pWPI/HA-Parp7, pWPI/Parp9 WT or mutants-Avi/GFP-BirA*, pLKO.1 GFP shRNA (shGFP, Addgene 30323), shParp7 (Sigma SHCLNG-NM_015508), pLKO.Tet-On (shControl, Novartis), Tet-shDtx3L with target sequence CGCGTATTAGGAGTCTCAGAT, pKH3 (Addgene 12555), pKH3/HA-Parp9, pKH3/Parp9, pBacPAK8 (Clontech 631402), pBac-PAK8/T7-His-Parp7, pET-Duet1/His6-Parp9/Dtx3L, pMBPHis-Parallel 1 plasmid

(MBP-His, Dr. Zygmunt Derewenda, University of Virginia), MBP-Parp9-MD1, MBP-Parp9-MD2, MBP-Parp9-MD1-MD2.

**Drug treatments**. The synthetic androgen R1881 (2 nM final) was used in most experiments, for treatment times that are specified in the figures. Other agonists and antagonists used were as follows: dihydrotestosterone (DHT; 10 nM), 4′-androstane-3,17-dione (ASD; 100 nM), Casodex (10 μM), flutamide (10 μM), hydroxyflutamide (HO-Flut; 100 nM), dehydroepiandrosterone (DHEA; 100 nM), estradiol (100 nM), cyproterone acetate (CPA; 100 nM), and enzalutamide (20 μM). In cycloheximide (CHX) experiments, cells were initially treated with R1881 for 4 h before adding CHX (100 μg/ml final concentration) and incubating for an additional 6 h. For AviTag labeling, PC3AR(pWPI/Parp9 WT or mutants-Avi/GFP-BirA*) cells were incubated with both R1881 (2 nM) and biotin (50 μM) for 18–24 h. The Parp inhibitors Veliparib and Olaparib were used at the concentrations shown in the figures. For inhibition of Parp1 automodification, Veliparib and Olaparib were added 30 min prior to hydrogen peroxide (0.88 mM final concentration) treatment. For the CHX and Olaparib chase experiment, PC3-AR were treated with R1881 (2 nM) overnight and followed with a chase by in the same medium containing R1881 (2 nM), CHX (100 μg/ml), and Olaparib (10 μM).

**Immunoprecipitation and Af1521 pulldown**. All steps were performed at 4 °C. Pelleted cells were resuspended in the cell extraction buffer (20 mM Tris-HCl pH 7.5, 100 mM NaCl, 0.5% Triton X-100, 1 mM PMSF, 2 mM DTT, 5 mM EDTA, 5 μg/ml each of aprotinin/leupeptin/pepstatin with optional 1 μM Olaparib), and then end-over-end rotated for 20 min. The extracts were clarified with centrifugation (16,800×$g$) for 20 min and then subjected to antibody beads (or GST-Af1521 beads) binding for 2.5–4 h. The beads were collected either by centrifugation for Agarose/Sepharose beads, or by magnetic fields for magnetic beads. The beads were washed five times with the wash buffer (20 mM Tris-HCl pH 7.5, 100 mM NaCl, 0.1% Triton X-100, 2 mM DTT, 0.1 mM EDTA, 1 μg/ml each of Aprotinin/Leupeptin/Pepstatin with optional 1 μM Olaparib), and then resuspended in SDS-loading buffer followed with western blot analyses. For silver staining, SDS-PAGE mini gels were incubated in the fixation solution (methanol: acetic acid: Milli Q water = 50:10:40) one time overnight, and the second time for 2 h. The gel was then washed 5 min each for five times with Milli Q water, incubated 90 s in the sodium thiosulfate solution (one pellet per 250 ml Milli Q water), rinsed three times with Milli Q water, incubated 20 min with silver nitrate solution (90 mg of AgNO₃ in 50 ml Milli Q water), rinsed again three times with Milli Q water, incubated in the developing solution (95 ml Milli Q water, 2 g of K₂CO₃, 5 ml above sodium thiosulfate solution, and 50 μl 37% formaldehyde) until reaching desired bands intensity, and then put into the quenching solution (10% acetic acid).

**ADP-ribosylation assays**. Parp7 ADP-ribosylation assays were performed at 30 °C for 30 min in 50 mM Tris-HCl (pH 7.5), 50 mM NaCl, 0.1 mM EDTA, 0.1% Triton X-100, 2 mM DTT containing ~15 ng/μl Parp7 enzyme along with either 1 mM NAD⁺ or 0.1 mM unlabeled NAD⁺ plus 0.025 mM biotin-NAD⁺. To test for AR ADP-ribosylation, the M2 beads carrying the immunoprecipitated Flag-AR were incubated with the assay solution, separated by centrifugation or magnet, washed with PBS + 0.1% Triton X-100. The proteins were subjected to SDS-PAGE followed by WB. ADP-ribosylation of AR on blots was detected with the Fl-Af1521 or fluorescently-labeled Neutravidin, and scanned and quantified on the Odyssey Infrared Imager (LICOR). Chemical release of ADPr from AR was performed using HgCl₂ in 100 mM Tris-HCl, pH 7.5 at room temperature for 1–2 h. AR was also incubated with 0.1 M hydroxyamine pH 7.5 or 0.1 M CHES buffer pH 9 at 37 °C for 2 h. Radioactive ADP-ribosylation assays measuring Parp7 activity were conducted in 50 mM HEPES, 5 mM β-Mercaptoethanol (BME), and 50 mM MgCl₂. ³²P-NAD⁺ was added (0.05 μM labeled plus 24.95 μM unlabeled NAD⁺) at 35 °C for 30 min. Autoradiography was used to detect auto-ADP-ribosylation. To measure and quantify ADP-ribosylation, proteins were captured on 0.2 μm nitrocellulose filters (GE Healthcare Life Sciences), washed, and ³²P signal was counted using a scintillation counter (Beckman). For assays that included inhibitors, complete reactions excluding NAD⁺ were pre-incubated on ice for 30 min. NADase (Sigma) solubilized by lipase was used to make a 1 mM ADP-ribose stock with unlabeled NAD⁺ and 1.2 μM radiolabeled NAD⁺. Generation of ADP-ribose by NADase at 37 °C for 30 min was confirmed by thin-layer chromatography. ADP-ribose was purified using a Microcon Centrifugal Filter (Millipore). Binding reactions were incubated at room temperature for 1 h and included 100 μM ADP-ribose and 4 μM recombinant MD protein. Proteins were captured on a 0.2-μm nitrocellulose filter (GE Healthcare Life Sciences), washed, and ³²P signal was counted using a scintillation counter (Beckman). ADP-ribosylated Parp1 was generated in reactions containing 50 mM Tris-HCl pH 7.5, 100 mM NaCl, 10 mM MgCl₂, 2 mM DTT, 2 μM PARP1C and 4 μM HPF1. The reaction was held on ice for 10 min, supplemented with 0.2 mM β-NAD⁺, further incubated on ice for 10 min, and finally supplemented with 2 μM Duplex oligo pair and incubated at 37 °C for 30 min. NAD⁺ was removed by desalting on a spin column (Thermo Scientific 89891), and aliquots were snap-frozen and stored at −80 °C. Dtx3L/Parp9 ubiquitylation/ADP-ribosylation was performed in 50 mM Tris-HCl (pH 7.5), 50 mM NaCl, 5 mM MgCl₂ buffer, containing 0.1 mM ATP, 1 mM DTT, 0.1 mg/ml T7His6-ubiquitin, 0.01 mg/ml each of E1 and E2, recombinant and IP'd Dtx3L/

His6-Parp9, β-NAD⁺ (12.5 μM biotin-NAD⁺ + 0.1 mM β-NAD⁺), and optional 1 μM Olaparib. For NUDT16 and ARH3 treatment, Flag peptide-eluted AR, immunoprecipitated AR and AR extract were incubated at 37 °C for 60 min with 3 μM NUDT16 in a buffer containing 50 mM Tris-HCl (pH 7.5), 50 mM NaCl, 5 mM MgCl₂, 2 mM DTT, 0.1% Triton X-100, 0.2 mg/ml BSA, 2 μM Olaparib and protease inhibitors. ADP-ribosylated PARP1C was incubated at 37 °C for 60 min with 0.1 μM ARH3 in a buffer containing 20–50 mM Tris-HCl (pH 7.5), 50 mM NaCl, 5 mM MgCl₂, 2 mM DTT, 2 μM Olaparib and 0.2 mg/ml BSA. For chemical sensitivity analysis, ADP-ribosylated PARP1C was incubated with 50 mM HgCl₂ or 100 mM NaCl in a buffer containing 100 mM Tris-HCl, pH 8 at room temperature for 60 min.

**Mass spectrometry**. For the first MS experiment examining ADP-ribosyl modification ($n = 1$), the sample was reduced with 10 mM DTT followed by alkylation with 50 mM iodoacetamide—each at room temperature for 1 h. In total, 0.5 μg of Promega alkylated trypsin was added to each tube for overnight digestion. The samples were quenched with 3 μL of acetic acid and reduced in volume to 20 μL. The samples were spun before loading to remove any possible precipitate. The LC-MS system consisted of a Thermo Electron Velos Pro Orbitrap mass spectrometer system with a Protana nanospray ion source interfaced to an in-house, self-packed 8 × 75 μm id (Phenomenex Jupiter 10 μm C18 reversed-phase) capillary column. 10 μL of the extract (technical and biological $n = 1$) was injected and the peptides eluted from the column by an acetonitrile/0.1 M acetic acid gradient at a flow rate of 0.5 μL/min over 1.3 h using an Agilent 1200 quaternary HPLC. The nanospray ion source was operated at 2.5 kV and the heated ion capillary set to 265 °C. The digest was analyzed using the rapid switching capability of the instrument acquiring a full scan mass spectrum to determine peptide molecular weights followed by product ion spectra (20) to determine amino acid sequence in sequential scans. The settings for these scans are as follows—MS collected in the Orbitrap (60 k resolution, 1 uscan, AGC 9E5, max ion time 500 ms), MS/MS CID collected in ion trap (1 uscan, AGC 8E3, max ion time 25 ms, min signal 1000, isolation 3, NCE 35, activation time 10). The instrument control software was set for dynamic exclusion—repeat 1, repeat duration 30, exclusion duration 60, list size 200. The data were analyzed by database searching using the Sequest algorithm within Thermo Proteome Discoverer 2.1 against human androgen receptor (SwissProt P10275). Search parameters—data extraction setting (precursor mass 350–5000, peptide length 5–144, total intensity 0, min peaks 1, FT S/N threshold 0), parent 10 ppm, fragment 0.7 Da, dynamic modifications (M oxidation, C carbamidomethyl, CDERK ADP ribosyl), 1 missed cleavage, and tryptic (KR). Peptides were filtered by charge state and xcorr— +1 > 1.8, +2 > 2.0, +3 > 2.2, +4 > 2.5. Peptides that passed these minimal criteria were manually examined by Dr. Sherman to first determine the modified peptide mass was within 5 ppm of predicted and second to determine the site of modification. Initial searches against common modification sites (E, D, R, K) yielded no obvious results. Four sites—C125, C284, C327, and C406—were identified after consideration of Cys as a potential modification target. Examination of all spectra showed low mass ions 348 (AMP), 428 (ADP), 542(ADP-ribose), and 524 (ADP-ribose—water).

For the second MS experiment examining phosphoribosyl modification ($n = 1$), NUDT16 was used to treat IP'd AR[27]. The sample was reduced with 10 mM DTT in 0.1 M ammonium bicarbonate, then alkylated with 50 mM iodoacetamide in 0.1 M ammonium bicarbonate (both room temperature for 0.5 h). The sample was then digested overnight at 37 °C with 0.1 μg (trypsin, chymotrypsin, Glu-C— 3 separate digests) in 50 mM ammonium bicarbonate. Phosphoribosylated peptides were enriched using a TiO column. The LC-MS system consisted of a Thermo Electron Q Exactive HF mass spectrometer system with an Easy Spray ion source connected to a Thermo 75 μm × 15 cm C18 Easy Spray column (E800). In total, 80% of the extract (each digest separate, technical and biological $n = 1$) was injected and the peptides eluted from the column by an acetonitrile/0.1 M formic acid gradient at a flow rate of 0.3 μL/min over 1.0 h using a Thermo nLC1200 HPLC. The nanospray ion source was operated at 1.9 kV, the heated capillary set to 250 °C, and the RF set to 40. The digest was analyzed using the rapid switching capability of the instrument acquiring a full scan mass spectrum to determine peptide molecular weights followed by product ion spectra (10) to determine amino acid sequence in sequential scans. The settings for these scans are as follows—MS collected in the Orbitrap (120 K resolution, 1 uscan, AGC 3E6, max ion time 60 ms), MS/MS HCD collected in Orbitrap (30 K resolution, 1 uscan, AGC 1E5, max ion time 60 ms, min signal 2E3, isolation 2, NCE 27). The instrument control software was set for automatic dynamic exclusion for 20 s and a lock mass of 445.12006. The data were analyzed by database searching using the Sequest search algorithm within Thermo Proteome Discoverer 2.4SP1 against human androgen receptor (SwissProt P10275). Search parameters—data extraction setting (precursor mass 350–5000, peptide length 5–144, total intensity 0, min peaks 1, FT S/N threshold 0), parent 10 ppm, fragment 0.02 Da, dynamic modifications (M oxidation, C carbamidomethyl, CDERK phosphoribosyl), no enzyme. Peptides were filtered by charge state and xcorr— +1 > 1.8, +2 > 2.0, +3 > 2.2, +4 > 2.5. Peptides that passed these minimal criteria were manually examined by Dr. Sherman to first determine the modified peptide mass was within 3 ppm of predicted and second to determine the site of modification. Initial searches against common modification sites (E, D, R, K) yielded no obvious results. Eight sites—

C131, C290, C406, C519, C596, C602, C620, and C670—were identified after consideration of Cys as a potential modification target.

MS/MS analysis of recombinant His6-T7-Parp7 ($n = 1$), which was carried out as described for AR but without NUDT16 treatment, and yielded 82% sequence coverage.

The mass spectrometry proteomics data have been deposited to the ProteomeXchange Consortium via the PRIDE (https://www.ebi.ac.uk/pride/archive/) partner repository with the dataset identified PXD018811 (AR ADP-ribosylation sites) and PXD025195 (Parp7 ADP-ribosylation sites).

**Recombinant proteins.** The pET-DUET1/His6-Parp9/Dtx3L plasmid was used for co-expression of human His6-Parp9 and untagged human Dtx3L in the bacterial strain *BL21(DE3)pLysS* with 0.4 mM IPTG at 18 °C overnight. MBP and MBP tagged proteins (cloned in the pMBPHis-Parallel 1 plasmid), GST, GST-Af1521, and GST-Af1521^tandem (Cloned in pGex-4T-2) were expressed in the bacterial strain *BL21* with 0.4 mM IPTG at 18 °C overnight. Parp7 was cloned into baculoviral transient plasmid pBacPAK8 with N-terminal His6-T7 tag. Plasmids were co-transfected with linearized BacPAK6 DNA (Clontech) to prepare the initial baculovirus, followed with virus amplification and then protein expression in insect cell line Sf21 at 27 °C for 5–6 days. The purification of His6-, GST-, and MBP-fusion proteins were done at 4 °C on TALON (Takara 635502), Glutathione (Sigma G4510) or Amylose beads (New England BioLabs E8021S). Fluorescent labeling of purified GST-Af1521^tandem was conducted using the IRDye 800CW Protein Labeling Kit (LICOR #928-38040).

**Macrodomain binding.** ADP-ribosylated AR was harvested from mammalian cells using cell lysis buffer (20 mM Tris-HCl pH 7.5, 50 mM NaCl, 0.5% Triton X-100, 5 mM EDTA, 2 mM DTT, and protease inhibitors) and clarified by centrifugation prior to incubation with recombinant Af1521 (10 µg GST-Af1521/10 µl Glutathione-agarose beads) or recombinant Parp9 MD fused to MBP. Binding reactions were supplemented with 0.5% Triton X-100, and mixed end-over-end for 4 h at 4 °C. The beads were washed five times with binding buffer, and fractions examined by immunoblotting with antibody.

**Immunofluorescence microscopy.** Cells were grown on coverslips, washed once with PBS, fixed with 3.75% formaldehyde for 10 min, washed three times with PBS, and permeabilized with 0.2% Triton X-100 for 5 min. Cells were washed three times with PBS and treated with blocking solution (2% BSA and 2% FBS in PBS) for 30 min. Primary and secondary antibodies were diluted in the blocking solution prior to incubation. The cells were incubated in primary antibodies for 2 h at RT, washed three times in PBS, incubated with secondary antibodies for 1 h at RT, incubated with DAPI solution in PBS (1 µg/ml) for 15 s, and mounted using Vectashield.

**RNA-seq.** Androgen-regulated gene expression[21] was analyzed using RNA-seq data from prostate cancer lines treated +R1881 for 0, 6, and 12 h (PC3-AR), and 0 and 24 h (VCaP) GSE120660 (Androgen effect on PC3-AR and VCaP cells). The effect of Dtx3L on androgen-regulated gene expression was performed in a similar manner in VCaP control (shVector, pLKO-Tet-ON) and Dtx3L-depleted (shDtx3L, pLKO-Tet-ON) cell lines (treatment +Dox for 4 days, followed by +R1881 for 24 h GSE133876 (Dtx3L and androgen effect in VCaP cells). RNA was prepared using a kit (Qiagen RNeasy) and library construction and sequencing performed by HudsonAlpha. RNA-seq data were analyzed using the Galaxy server (https://usegalaxy.org/). Transcript quantification was performed using Salmon[58] to map to the hg38 human genome build, and DESeq2 within the Galaxy site was used for normalizing count data, estimating dispersion, fitting a negative binomial model for each gene, and comparing expression between groups[59]. A fold cutoff of +/− 0.5 log2 and an adjusted *P* value of <0.001 was considered significant. Distributions of genes within overlapping gene sets were compared using a $2 \times 2$ contingency table and Chi-squared test[60], and the degree of overlap compared by Fisher exact or Chi-squared test.

**Gene expression and reporter assays.** For real-time quantitative PCR (RT-qPCR) experiments, RNA was extracted using TRIzol (Thermo Fisher Scientific) or the Qiagen RNeasy kit. The cDNA synthesis was carried out using Bio-Rad iScript reagents. DNA amplification using primer pairs (Supplementary Fig. 3) was done in the presence of the SensiMix Sybrgreen reagents, all according to manufacturer instructions. Technical replicates were averaged and normalized to housekeeping genes *GUS*. The results shown are representative of at least three experiments. PSA-luciferase experiments in Parp9 knockout cells were performed using a dual-luciferase reporter assay system (Promega), making use of the complete 6.1-kB promoter for the KLK3 gene (PSA). This involved co-transfecting plasmids for PSA promoter-firefly luciferase, cytomegalovirus promoter-*Renilla* luciferase, and either WT Parp9 or mutant Parp9 (G112,311E). Assays were performed in triplicate and the data plotted as firefly:Renilla. Immunoblotting for Parp9 was used to establish protein expression levels.

**Chromatin immunoprecipitation (ChIP).** Cells were fixed using 1% formaldehyde in growth medium at 37 °C for 10 min and neutralized with 125 mM glycine at room temperature for 5 min. Cells were harvested in cold PBS + 1 mM PMSF at 4 °C by centrifugation (2200 × g, 4 °C, 5 min), resuspended in the extraction buffer (20 mM Tris pH 7.5, 1 mM EDTA, 100 mM NaCl, 2 mM DTT, 0.5% Triton X-100, and protease inhibitors) with end-over-end rotation at 4 °C for 20 min. Pellets were collected by centrifugation (2200 × g, 4 °C, 5 min), washed with EDTA-free extraction buffer to remove the EDTA, and then sonicated (Scale 4, 50% cycle, 10 pulse; repeat two more times). Samples were supplemented with CaCl₂ (4 mM final) and micrococcal nuclease (NEB 500 gel units) to cleave the DNA at 37 °C for 5 min. EGTA (50 mM final) and SDS (0.1%) were added to quench the nuclease reaction. The samples were incubated for 10 min at 4 °C, followed by centrifugation (16,800 × g, 4 °C, 20 min). The supernatant was pre-absorbed with unconjugated Protein G beads in the presence of BSA (1 mg/ml final) and salmon sperm DNA (0.2 mg/ml final), then subjected to antibody bead binding at 4 °C for overnight, with rabbit IgG beads as the negative control. Beads were collected with centrifugation, washed five times with the extraction buffer supplemented with 0.1% SDS, then resuspended in 100 µl of reverse cross-linking buffer (125 mM Tris pH 6.8, 5% BME, and 1% SDS), and incubated at 65 °C for 6 h. DNA was purified using the QIAquick Gel Extraction kit (Qiagen). Standard SensiMix Sybrgreen-based reactions were used to quantify the DNA.

**Af1521 detection of chromatin-associated ADP-ribosylation (ChAP).** Cells were washed once with PBS, fixed at room temperature with 3.7% formaldehyde and 1 µM Olaparib in PBS for 5 min, washed twice with cold (4 °C) TBS (20 mM Tris-HCl pH 7.5 and 100 mM NaCl), permeabilized with cold methanol for 10 min, washed twice with cold TBST (TBS and 0.15% Tween-20), and then harvested in cold TBST and 1 µM Olaparib by centrifugation (2200 × g, 4 °C, 5 min). Cells were resuspended in the extraction buffer (TBS, 0.5% Triton X-100, 5 µg/ml each of aprotinin/leupeptin/pepstatin, 1 mM PMSF, 2 mM DTT, and 1 µM Olaparib; 400 µl for a plate of cells), and sonicated on ice for three times with 10 pulse each at scale "4" with 50% cycle for each time. Samples were supplemented with CaCl₂ (4 mM final) and DNA was cleaved using Micrococcal Nuclease (500 gel units) for 5 min at 37 °C. After adding EGTA (50 mM final) and SDS (0.03% final), the mixture was incubated at 4 °C for 10 min, and then centrifugated at 4 °C for 20 min at 16,800 × g. The supernatant was transferred to a fresh siliconized tube with 5 µl of packed Glutathione MagBeads, supplemented with BSA (1 mg/ml final) and salmon sperm DNA (0.2 mg/ml), and incubated at 4 °C for 2 h. The solution was separated from the magnetic beads; 10 µl of the clarified solution was set aside as 5% input, and 200 µl was incubated at 4 °C for 3 h in another siliconized tube with the pre-formed beads (4 µg GST/2 µl packed Glutathione MagBeads; 4 µg GST-AF1521^tandem/2 µl packed Glutathione MagBeads; or 2 µl packed M2-magnetic beads). The magnetic beads were separated from the solution, washed 5 times with the wash buffer (TBS, 0.1% Triton X-100, 1 µg/ml Aprotinin/Leupeptin/Pepstatin, 2 mM DTT, 0.1 mM EDTA, 0.03% SDS, and 1 µM Olaparib), and incubated at 65 °C for 6 h in the reverse cross-linking buffer (125 mM Tris pH 6.8, 5% BME, and 1% SDS). DNA was purified using the QIAquick Gel Extraction kit (Qiagen). Standard SensiMix Sybrgreen-based reactions were used to quantify the DNA. For re-ChIP assays, the GST-AF1521^tandem/glutathione MagBeads with associated DNA were eluted with 100 µl of wash buffer supplemented with 1 mg/ml BSA, 0.2 mg/ml salmon sperm DNA and 0.5 mM ADPr, and the eluted material was further immunoprecipitated with 2 µl of M2-magnetic beads, followed with five times washing with wash buffer. The M2-magnetic beads pre-incubated with 5 µg of Flag peptide were used as a negative control.

**Additional data analysis.** In silico protein modeling was performed in PyMol by aligning 5AIL (Parp9 MD2) with 2BFQ (Af1521 structure) and 3VFQ (Parp14 MD2). In addition, Parp9 MD1 primary sequence was threaded onto 3VFQ (Parp14 MD2 structure) using SwissModel. Further refinement was achieved using PyMol. ADP-ribosylated peptide sequences (14 total) from AR and Parp7 were analyzed using Seq2Logo[34]. PRAD (prostate adenocarcinoma) data for box and whisker plots was downloaded from the publicly available TCGA Data Portal (http://cancergenome.nih.gov/tcga). RSEM (RNA-Seq by Expectation-Maximization) values were used as relative expression. Significance was calculated using a paired *t* test. Similarly, using another publicly available dataset from Memorial Sloan Kettering Cancer Center (MSKCC)[42,43], Parp7 mRNA was assessed in prostate cancer metastases and primary tumors. For disease recurrence, MSKCC prostate cancer patient data[61] was stratified by Parp7 mRNA expression levels and plotted using cBioPortal. Details for the significance tests can be found within the text and was performed using GraphPad Prism6.

**Statistics and reproducibility.** The MS/MS discovery of ADP-ribosylation sites was performed once, and validated by mutagenesis and chemical sensitivity analysis. RNA-seq and reporter gene assays were both performed as replicates ($n = 3$ biological independent experiments). All other experiments were performed as replicates ($n = 2$ biologically independent experiments) with similar results.

**Reporting summary**. Further information on research design is available in the Nature Research Reporting Summary linked to this article.

## Data availability

We will make freely available any datasets necessary to interpret and replicate the methods and findings reported in this article. RNA-seq data generated in our studies are publically available at GEO under the accession codes GSE120660 (Androgen effect on PC3-AR and VCaP cells) and GSE133876 (Dtx3L and androgen effect in VCaP cells) AR ChIP-seq data were previously published[31,32] and are available at GEO under the accession codes GSE28126 (Androgen effect on AR ChIP-seq in VCaP cells) and GSE54202 (Androgen effect on AR ChIP-seq in PC3-AR cells). The mass spectrometry proteomics data have been deposited to the ProteomeXchange Consortium via the PRIDE (https://www.ebi.ac.uk/pride/archive/) partner repository with the dataset identifiers PXD018811 (AR ADP-ribosylation sites) and PXD025195 (Parp7 ADP-ribosylation sites). Protein structure data were previously published by others with the PDB codes 5AIL (Parp9 MD2), 2BFQ (Af1521 structure) and 3VFQ (Parp14 MD2). Source data are provided with this paper.

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

## Acknowledgements

These studies were supported by NCI award R01CA214872 to B.M.P. and NIH award R01GM135376 to T.A. We thank Mr. Adam Spencer for technical assistance and Drs. Anthony Leung (Johns Hopkins University) and Mike Guertin (University of Virginia) for helpful discussions.

## Author contributions

Conceptualization: C.S.Y., K.J., T.K., L.O., D.W., and B.M.P.; investigation: C.S.Y., K.J., T.K., N.D., L.O., B.R., Y.P., I.K., N.E.S., K.D., T.A. N.E.S., D.W., and B.M.P.; writing—original draft: C.S.Y., K.J., L.O., D.W., and B.M.P.; writing—review & editing: C.S.Y., K.J., T.K., L.O., I.K., N.E.S., D.W., and B.M.P.; funding acquisition: B.M.P.; resources: C.S.Y., K.J., T.K., L.O., I.K., N.E.S., D.W., and B.M.P.; supervision, D.W. and B.M.P.

## Competing interests

The authors declare no competing interests.
