## [Peer Review File · Nature Communications]

REVIEWER COMMENTS

Reviewer #1 (Remarks to the Author):

Yang et al. present a set of experiments to describe the role of PARP7 (ADPr writer) and PARP9/Dtx3L (ADPr reader) in androgen signalling and identify by mass spectrometry Cys-ADPr on AR. Overall the study is of interest to the broad readership of Nature Communications, but I have a few concerns that should be addressed.

1. Fig.2G and 2H

Although I appreciate authors' efforts to identify the hydrolase responsible for the reversal of AR ADPr, I am not sure what we can conclude from "ARH3 showed the highest activity releasing ~40% of the ADP-ribose from AR"? Do the authors imply that Cys-ADPr on AR is reversed by ARH3? If so, they should be able to find assay conditions that completely remove the signal as they did with the complete removal with the exposure to HgCl₂. If ARH3 removes non-Cys-ADPr, the authors should be able to find by MS this non-Cys-ADPr on AR. This is perhaps due to the fact that they do not include Ser, which is efficiently and completely removed by ARH3, when analyzing their data by database searching.

2. Identification of Cys-ADPr sites by mass spectrometry

This part would have benefited from the use of fragmentation techniques that are better suited to ADPr, which is a very labile PTM, namely high-resolution ETD (or a combination of ETD with HCD or infrared photoactivation). Also, it is not clear why they use a Velos Orbitrap (I guess low resolution CID although I cannot find this information in Materials & Methods) for ADPr and a Q Exactive for pRib peptides. It would have been better to use high resolution HCD for all the samples. Nevertheless, Cys-ADPr appears less labile than some other forms of ADPr and the absence of alkylation indicate that Cys are indeed the modified residues. However, the use of ETD might have led to the identification of other forms of ADPr that might have explained the partial removal by ARH3.

3. fig 5

The authors show that olaparib inhibits AR ADPr. As olaparib is mainly a PARP1/2 inhibitor with some activity towards PARP7, I would be much more confident about the dependence of AR ADPr on PARP7 if the authors could confirm these data with a genetic knock-out of PARP7.

4. fig 6

The data in Fig6 doesn't directly support the hypothesis that the Dtx3L/PARP9/AR complex can affect transcription. Rather, the authors show that Dtx3L and PARP7 by themselves can, but this effect may be independent of formation of the complex. At minimum, they should show if PARP9 KO can also downregulate a relevant subset of genes. However, a more direct test of the Dxt3L/PARP9 complex hypothesis should be performed. For example, by mutating AR target cysteines to prevent ADPr-dependent complex formation and assess the effect on transcription upon R1881 treatment. Alternatively, by using PARP9 lacking MD, or by using the G112E and G311E that they characterize as abolishing the complex formation.

5. fig 7

The authors could add, as control for pulldown in ChIP settings, an experiment where the tandem ADPr ChAP-AR ChIP pulldown is performed, and the eluted proteins are run on WB and blotted against Dxt3L, PARP9, and AR (as control). Conceptually similar to Fig.4E but in the context of ChIP and with an additional macrodomain IP.

It is a bit problematic that according to Fig.7F, olaparib treatment doesn't affect AR binding to MYBPC1 site2. This seems to imply that if ADPr-dependent Dxt3L/PARP9/AR complex is indeed binding to MYBPC1, it's contribution is only an undetectable fraction relative to pure AR binding to MYBPC1 and calls into question the overall relevance of the results.

Furthermore, Fig.7H shows olaparib-dependent decrease in MYBPC1 expression which, combined with the results of 7F, seems to imply that the olaparib effect is actually independent of the Dxt3L/PARP9/AR complex. One possibility, according to Fig6I, is that PARP7 may have by itself (independently from Dxt3L/PARP9) an ADPr-dependent effect on MYBPC1 transcription, which would confound olaparib experiments.

Reviewer #2 (Remarks to the Author):

Yang and colleagues describe a beautiful molecular system where activated androgen receptor (AR) induces PARP7 expression, which mono-ADPrylates AR on cysteine residues that is read by the macrodomains of PARP9 in the PARP9/Dtx3L complex. Authors also study the effect of Dtx3L knockdown on gene regulation by AR. They detect ADPrylated AR at the promoter of MYBPC1, a gene that they show is regulated by PARP7. The manuscript is well written, the experiments are properly done. I think this study would be of interest to the general readership of the journal.

How long is AR mono-ADPrylated after its inactivation?

In addition, does the mono-ADPrylation of AR by PARP7 affect AR activity/targets independent of PARP9/Dtx3L? Are there genes that are regulated by ADPrylated AR and not by its subsequent binding of PARP9/Dtx3L? ADPrylated AR could potentially be targeted to chromatin enriched for macroH2A1.1 that is likely to bind ADPrylated AR.

How does the binding of AR by PARP9/Dtx3L impact its activity? Is AR ubiquitinated upon PARP9/Dtx3L binding? Can authors detect histone ubiquitination by PARP9/Dtx3L in the vicinity of the studied MYBPC1 promoter? Do authors have any data whether the activity of PARP9 – ADPrylation of ubiquitin – impacts AR-regulated gene expression?

Minor comment:

The observation that olaparib inhibits PARP7 and the association of PARP7 with ovarian cancer survival – lines 312-319 – is a very interesting and important corollary, however, I would consider moving it to the discussion as it breaks the line of thought of the study and it might gain more impact – which it deserves – when the overall findings and potential ramifications of the study are discussed.

Reviewer #3 (Remarks to the Author):

Review of NCOMMS-20-14795 by Yang et al. (Paschal)

“An androgen signaling axis uses a writer and a reader of ADP-ribosylation to control protein complex assembly”

Brief summary

In this manuscript, the authors sought to characterize how ADP-ribosylation – a posttranslational modification of proteins – influences androgen receptor (AR) activity. The authors focused on the role of the mono-ADP-ribosyl transferase, Parp7, in androgen-regulated AR activity and responses. They showed that (1) Parp7 ADP-ribosylates AR on multiple cysteine residues, (2) this modification is recognized by the ADP-ribose-binding macrodomains in Parp-9, and (3) is necessary for the recruitment of Dtx3L/PARP9, a heterodimeric complex with E3 monoubiquitylation activity, which helps to regulate a subset of AR-regulated genes. Further, based on these insights, the authors examined the effects of treatment with Olaparib, a Parp1/2 inhibitor, on this pathway. . The authors conclude that they have identified “components of a new androgen signaling axis that uses a writer [Parp-7] and reader [Parp-9] of ADP-ribosylation to modulate AR activity.”

Strengths and weaknesses

This focus of this paper on mono-ADP-ribosylation, a mono-ADP-ribosyl transferase (i.e., Parp7) and a reader of ADP-ribosylation (i.e., Parp9) are of interest because these are poorly understood aspects of ADP-ribosylation and the PARP family of enzymes. The specific molecular mechanisms revealed in this study are unique and, if true, will add more broadly to the understanding of ADP-ribosylation, PARPs, and the functions of macrodomain readers.

Strengths: The authors have analyzed Parp7-mediated regulation of the AR pathway, adding a new level of understanding and connectivity between ADP-ribosylation and steroid hormone signaling. They explored Parp7 mediated ADP-ribosylation of AR, including the potential identification of sites of ADP-ribosylation and the formation of an AR-Dtx3L/Parp9 complex that contributes to the regulation of a subset of AR-responsive genes.

Weakness: The evidence for the biological consequences of AR ADP-ribosylation in prostate cancer and the necessity of this modification for the function of AR is relatively weak. Furthermore, the mechanism of action is not clearly delineated and could be strengthened by some additional experiments. Finally, the approaches used for identification and proof of the sites of ADP-ribosylation on AR leave room for doubt.

Specific criticisms

Major comments:

- (1) The figure legends lack necessary details that would help explain the exact nature of each experiment, including the number of replicates and statistics, and not just the conclusion drawn from the figure. In particular, blots that show quantification should have statistics.
- (2) In the text, the authors note that Dtx3L and Parp9 are ~85 kDa, but the Western blots in Figure 1A shows their size to be over 98 kDa. Why is there a discrepancy?
- (3) In which cellular compartment is AR ADP-ribosylated? Parp9/DTX3L is localized to the cytoplasm, whereas Parp7 is known to translocate between the nucleus and cytosol. Once AR is activated by its ligand, it localizes to the nucleus, so where does the ADP-ribosylation and complex formation occur. Does this have any effect on the translocation of AR to the nucleus?
- (4) In Figure 2B, why is NAD⁺ supplementation in the medium required for the ADP-ribosylation of AR? The NAD⁺ permeability of cells is debated in the literature and ecto-enzymes such as CD38 may degrade it. Does the same effect occur with NMN, which is freely cell permeable?
- (5) In Figure 3, the authors should show Parp7 levels, both after knockdown and over expression.
- (6) Do the G112E and G311E mutations in the Parp9 macrodomains shown in Figure 4D actually cause loss of binding to ADP-ribose? Although modeled on Parp14 mutations, this should be tested independently.
- (7) Most of the sites of AR ADP-ribosylation identified in Table 1 and Figure 4 are in the NTD, which forms activation function 1 (AF1). Yet, the authors show that ligand binding (and presumably the formation of an active AF2) is required for ADP-ribosylation of AR. While deleting the entire domains in AR (e.g., Figure 3K) results in the loss of ADP-ribosylation, this can be due to overall effects on protein structure. These might be more clearly resolved using specific AR point mutants that prevent ligand and/or DNA binding.
- (8) Figure 4I is confusing, but if I am interpreting the figure correctly, it does not appear to show a loss of AR ADP-ribosylation with the site mutants, except for the extremely mutated 11CG mutant, which may have severe structural defects.
- (9) Also in Figure 4I, why are both HA-tagged and Flag-tagged AR required in this experiment? The figure should be labeled more intuitively and the experiment should be explained more clearly. It's difficult to interpret, but mutating these sites seems to increase the mobility of the AR, perhaps indicating alterations to other PTMs. Overall, this experiment leaves one wondering if these cysteines are not the actual sites of ADP-ribosylation in AR.
- (10) Figure 6, A-G shows the effect of loss of DTX3L on AR-dependent gene expression. The number of genes affected by the loss of DTX3L are less than 10% of the AR transcriptome. Furthermore, from the analysis shown, it is not entirely clear what the function of DTX3L is in regulating AR-dependent transcription. Perhaps, having browser tracks showing examples of the different modes of regulation would help the reader understand better. Additionally, what are the genes that are regulated in this manner and what is their importance in prostate cancer? With this level of analyses and explanation, the RNA-seq analysis contributes minimally toward understanding the mechanism or function of the AR:DTX3L/PARP-9 interaction. Perhaps examining the overlap between genes regulated by AR, DTX3L and Parp7 would help to bring these observations together.

Minor comments:

- (1) Af1521 Gly42Glu mutant that eliminates AR binding should be shown in Figure 2.
- (2) The blots that have numbers on the left side should make it clear what the numbers represent (e.g., Figures 1A and 3K).
- (3) Exposure of some blots make it difficult to see if there is a band present (e.g., Figures 2D and 4E).
- (4) The possibility of Parp9 macrodomain hydrolase activity was not discussed.

RESPONSES TO REVIEWERS NCOMMS-20-14795

Reviewer 1

1. Although I appreciate authors' efforts to identify the hydrolase responsible for the reversal of AR ADPr, I am not sure what we can conclude from "ARH3 showed the highest activity releasing ~40% of the ADP-ribose from AR"?

Response. We totally agree with the reviewer that treating an ADP-ribosylated substrate (in this case, AR) with recombinant hydrolases is not itself a very informative experiment. This is really a "first" experiment that requires extensive cell-based follow-up, and probably an entire study to establish the biological significance. For these reasons, we removed the hydrolase experiment. In our hands, recombinant ARH3 hydrolyzes ADP-ribose (from a Parp1 fragment) at an apparent half-maximal concentration of 10 nM. At this concentration, ARH3 has a very small effect on AR ADP-ribosylation. Whether this reflects the fact that ARH3 has low activity towards ADPr-Cys (which, to our knowledge, has not been directly tested), or whether it is suggestive of ADP-ribosylation on Ser sites in AR, is unclear. We acknowledge in the paper that we could have missed some sites by MS/MS (a persistent caveat of MS/MS). But, it is fair to point out that ADP-ribose is removed from AR by mercury treatment, and AR ADP-ribosylation was eliminated by mutating cysteines in AR. Additionally, the fact that AR with multi-site Cys mutations undergoes androgen-induced import (Supplementary Fig. 4B) indicates that ligand binding and an induced conformation sufficient for selective recognition by the nuclear import machinery is still achieved in the Cys-Gly substituted forms of AR.

2. Identification of Cys-ADPr sites by mass spectrometry

This part would have benefited from the use of fragmentation techniques that are better suited to ADPr, which is a very labile PTM, namely high-resolution ETD (or a combination of ETD with HCD or infrared photoactivation). Also, it is not clear why they use a Velos Orbitrap (I guess low resolution CID although I cannot find this information in Materials & Methods) for ADPr and a Q Exactive for pRib peptides. It would have been better to use high resolution HCD for all the samples. Nevertheless, Cys-ADPr appears less labile than some other forms of ADPr and the absence of alkylation indicate that Cys are indeed the modified residues. However, the use of ETD might have led to the identification of other forms of ADPr that might have explained the partial removal by ARH3.

Response. We agree that ETD could potentially identify more sites, but ETD instrumentation is not available at our institution. We did employ more than one type of MS instrumentation for the project based on what was available at our institution during the course of the work. The mutational analysis and the mercury sensitivity of ADP-ribosylation are consistent with the MS/MS results.

3. fig 5. The authors show that olaparib inhibits AR ADPr. As olaparib is mainly a PARP1/2 inhibitor with some activity towards PARP7, I would be much more confident about the dependence of AR ADPr on PARP7 if the authors could confirm these data with a genetic knock-out of PARP7.

Response. We agree that Olaparib has much lower activity towards Parp7, as indicated by the Ki we measured in Fig. 5F (1.1 μ M). While we do not have a Parp7 knockout line, we do show that efficient Parp7 depletion in two different prostate cancer cell lines (each with stable shRNA

to Parp7) causes a striking reduction in androgen-induced ADP-ribosylation (Fig. 3). This result is complemented by multiple experiments where ectopic expression of Parp7 promotes ADP-ribosylation. Parp7 also ADP-ribosylates AR *in vitro* at the sites necessary for Dtx3L/Parp9 binding, and this *in vitro* ADP-ribosylation by Parp7 is reversible with mercury (Fig. 3).

4. The data in Fig6 doesn't directly support the hypothesis that the Dtx3L/PARP9/AR complex can affect transcription. Rather, the authors show that Dtx3L and PARP7 by themselves can, but this effect may be independent of formation of the complex. At minimum, they should show if PARP9 KO can also downregulate a relevant subset of genes. However, a more direct test of the Dxt3L/PARP9 complex hypothesis should be performed. For example, by mutating AR target cysteines to prevent ADPr-dependent complex formation and assess the effect on transcription upon R1881 treatment. Alternatively, by using PARP9 lacking MD, or by using the G112E and G311E that they characterize as abolishing the complex formation.

Response. We agree there are probably multiple ways that Dtx3L/Parp9 and Parp7 could contribute to transcription, and complex formation with AR is just one of them. In terms of testing the basic model, we have added new data - conceptually similar to what is suggested by the reviewer –supporting the model that complex formation can promote AR-dependent transcription. Using Parp9 knockout cells, we show that AR and androgen-dependent transcription from the 6.1 kB PSA promoter (widely used in prostate cancer labs that study AR) is increased significantly by re-introduction of WT Parp9, but the effect is lost when the Parp9 macrodomains are mutated (G112E, G311E). This result (Fig. 6M) links the ADP-ribose binding activity of the Dtx3L/Parp9 complex to androgen-induced AR-dependent transcription.

5. fig 7. The authors could add, as control for pulldown in ChIP settings, an experiment where the tandem ADPr ChAP-AR ChIP pulldown is performed, and the eluted proteins are run on WB and blotted against Dxt3L, PARP9, and AR (as control). Conceptually similar to Fig.4E but in the context of ChIP and with an additional macrodomain IP.

Response. We agree that additional manipulations of the ChAP-AR-ChIP protocol could be informative, but the available antibodies to Dtx3L and Parp9 are not sensitive enough to detect proteins in ChIP samples.

It is a bit problematic that according to Fig.7F, olaparib treatment doesn't affect AR binding to MYBPC1 site2. This seems to imply that if ADPr-dependent Dxt3L/PARP9/AR complex is indeed binding to MYBPC1, it's contribution is only an undetectable fraction relative to pure AR binding to MYBPC1 and calls into question the overall relevance of the results.

Response. There is no reason a priori to think that ADP-ribosylation impacts the chromatin binding activity of AR, so we were not surprised that Olaparib had no effect on AR ChIP. In new data, we show that AR-dependent transcription from the PSA promoter is activated by Dtx3L/Parp9, and that this is dependent on the macrodomains of Parp9 (Fig. 6M).

Furthermore, Fig.7H shows olaparib-dependent decrease in MYBPC1 expression which, combined with the results of 7F, seems to imply that the olaparib effect is actually independent of the Dxt3L/PARP9/AR complex. One possibility, according to Fig6I, is that PARP7 may have by itself (independently from Dxt3L/PARP9) an ADPr-dependent effect on MYBCP1 transcription, which would confound olaparib experiments.

Response. We agree that Parp7 may have independent effects on transcription, which is now stated in the manuscript. Overall, the data shows that AR is ADP-ribosylated, Parp7 is the

responsible enzyme, the modification is necessary for complex formation with Dtx3L/Parp9, the macrodomains of Parp9 are required to bind modified AR, and complex formation enhances androgen and AR-dependent transcription from the well-defined PSA promoter.

Reviewer 2

How long is AR mono-ADP-ribosylated after its inactivation?

Response. We queried the stability of ADP-ribosylation on AR using an Olaparib-cycloheximide chase experiment (Fig. 5G). The data indicate that about half of the ADP-ribose is lost from AR on a timescale of ~60 minutes, while the remaining ADP-ribose is lost very slowly. These data contrast with the half-life of AR, which is >10 hrs.

In addition, does the mono-ADP-ribosylation of AR by PARP7 affect AR activity/targets independent of PARP9/Dtx3L? Are there genes that are regulated by ADP-ribosylated AR and not by its subsequent binding of PARP9/Dtx3L? ADP-ribosylation of AR could potentially be targeted to chromatin enriched for macroH2A1.1 that is likely to bind ADP-ribosylated AR.

Response. These are all interesting and relevant questions that require additional RNA-seq experiments done in various combinations (gene knockouts, drug treatments). We hope to tackle these questions in the future.

How does the binding of AR by PARP9/Dtx3L impact its activity? Is AR ubiquitinated upon PARP9/Dtx3L binding? Can authors detect histone ubiquitination by PARP9/Dtx3L in the vicinity of the studied MYBPC1 promoter? Do authors have any data whether the activity of PARP9 – ADP-ribosylation of ubiquitin – impacts AR-regulated gene expression?

Response. We were able to address the key question of how Dtx3L/Parp9 can affect AR activity using a PSA reporter assay in the background of Parp9 knockout cells. Using Parp9 knockout cells, we show that AR and androgen-dependent transcription from the 6.1 kB PSA promoter is increased significantly by re-introduction of WT Parp9, but the effect is lost when the Parp9 macrodomains are mutated (G112E, G311E). This result (Fig. 6M) links the ADP-ribose binding activity of the Dtx3L/Parp9 complex to androgen-induced AR-dependent transcription.

Minor comment:

The observation that olaparib inhibits PARP7 and the association of PARP7 with ovarian cancer survival – lines 312-319 – is a very interesting and important corollary, however, I would consider moving it to the discussion as it breaks the line of thought of the study and it might gain more impact – which it deserves – when the overall findings and potential ramifications of the study are discussed.

Response. We agree that the Olaparib effect on Parp7 and the Parp7 expression data in the context of patient survival is provocative and we highlight this in the manuscript.

Reviewer 3

(1) The figure legends lack necessary details that would help explain the exact nature of each experiment, including the number of replicates and statistics, and not just the conclusion drawn from the figure. In particular, blots that show quantification should have statistics.

Response. We apologize for the brevity of the figure legends, which have been expanded significantly to improve understanding. We also added many more details to the Methods. We removed the numbers associated with the blots since the effects are clear from the images.

(2) In the text, the authors note that Dtx3L and Parp9 are ~85 kDa, but the Western blots in Figure 1A shows their size to be over 98 kDa. Why is there a discrepancy?

Response. The molecular weight designations on the silver -stained gel were corrected.

(3) In which cellular compartment is AR ADP-ribosylated? Parp9/DTX3L is localized to the cytoplasm, whereas Parp7 is known to translocate between the nucleus and cytosol. Once AR is activated by its ligand, it localizes to the nucleus, so where does the ADP-ribosylation and complex formation occur. Does this have any effect on the translocation of AR to the nucleus?

Response. We addressed this interesting question by engineering AR with well-characterized transport signals that force its localization to the nucleus (SV40 NLS) or cytoplasm (cAbl NES) in an androgen-independent manner. These new data (Fig. 4F,G, H; Fig. S3B) show convincingly that ADP-ribosylation and AR-Dtx3L/Parp9 complex assembly occur in the nucleus. These experiments were performed in a background of WT AR (HA-tagged), where the mutants were Flag-tagged. This was done to ensure proper induction of endogenous Parp7, and it provided an internal control that allowed a careful comparison of ADP-ribosylation and complex formation of mutant and WT AR. In other words, we detected ADP-ribosylation and complex formation of WT and mutant AR from the same cells by immunoprecipitation via different tags.

(4) In Figure 2B, why is NAD⁺ supplementation in the medium required for the ADP-ribosylation of AR? The NAD⁺ permeability of cells is debated in the literature and ecto-enzymes such as CD38 may degrade it. Does the same effect occur with NMN, which is freely cell permeable?

Response. We did not use NAD⁺ supplementation in cell media for any experiments. We did use biotin-NAD⁺ in Fig. 1D but this was in the context of biochemical assays with purified proteins. We have not explored the use of NMN in cell-based assays.

(5) In Figure 3, the authors should show Parp7 levels, both after knockdown and over expression.

Response. Immunoblots of Parp7 knockdowns are shown in PC3-AR and VCaP cells (Supplementary Fig. 6).

(6) Do the G112E and G311E mutations in the Parp9 macrodomains shown in Figure 4D actually cause loss of binding to ADP-ribose? Although modeled on Parp14 mutations, this should be tested independently.

Response. We showed in a previous study (Yang et al. Mol Cell 2017) that the G112E/G311E mutations in Parp9 reduce binding to PAR in vitro. This paper is now cited in the text.

(7) Most of the sites of AR ADP-ribosylation identified in Table 1 and Figure 4 are in the NTD, which forms activation function 1 (AF1). Yet, the authors show that ligand binding (and presumably the formation of an active AF2) is required for ADP-ribosylation of AR. While deleting the entire domains in AR (e.g., Fig. 3K) results in the loss of ADP-ribosylation, this can

be due to overall effects on protein structure. These might be more clearly resolved using specific AR point mutants that prevent ligand and/or DNA binding.

Response. We agree that this is a potential caveat, and we performed the experiment requested by the reviewer. We engineered a point mutation in the LBD (based on a patient with androgen insensitivity syndrome), and tested the AR mutant for ADP-ribosylation and complex formation (Fig. 2G). We found that the LBD mutation reduced ADP-ribosylation and complex formation. This result links androgen-induced ADP-ribosylation to complex formation.

(8) Figure 4I is confusing, but if I am interpreting the figure correctly, it does not appear to show a loss of AR ADP-ribosylation with the site mutants, except for the extremely mutated 11CG mutant, which may have severe structural defects.

Response. The data in Fig. 4E show that 11 Cys residues have to be mutated to eliminate ADP-ribosylation, but complex formation can be abolished by mutating 8 Cys residues. This suggests that a subset of the ADP-ribosylation sites are used for complex formation.

(9) Also in Figure 4I, why are both HA-tagged and Flag-tagged AR required in this experiment? The figure should be labeled more intuitively and the experiment should be explained more clearly. It's difficult to interpret, but mutating these sites seems to increase the mobility of the AR, perhaps indicating alterations to other PTMs. Overall, this experiment leaves one wondering if these cysteines are not the actual sites of ADP-ribosylation in AR.

Response. WT AR was co-expressed with the mutant AR to ensure Parp7 induction, since endogenous Parp7 is performing the ADP-ribosylation. The rationale for co-expressing WT AR is now better articulated in the text.

(10) Figure 6, A-G shows the effect of loss of DTX3L on AR-dependent gene expression. The number of genes affected by the loss of DTX3L are less than 10% of the AR transcriptome. Furthermore, from the analysis shown, it is not entirely clear what the function of DTX3L is in regulating AR-dependent transcription. Perhaps, having browser tracks showing examples of the different modes of regulation would help the reader understand better. Additionally, what are the genes that are regulated in this manner and what is their importance in prostate cancer? With this level of analyses and explanation, the RNA-seq analysis contributes minimally toward understanding the mechanism or function of the AR:DTX3L/PARP-9 interaction. Perhaps examining the overlap between genes regulated by AR, DTX3L and Parp7 would help to bring these observations together.

Response. The major contribution of the RNA-seq analysis is that it provides a relatively unbiased test of whether Dtx3L affects AR-regulated genes. The new luciferase experiment complements the RNA-seq analysis, since in the latter, activation of the PSA gene was dependent on Dtx3L levels. Using the PSA promoter-luciferase, we showed that AR-dependent transcription was dependent on the macrodomain binding function of the Dtx3L/Parp9 complex. Also, the source data for Fig.6 contains a gene list that shows the R1881-Dtx3L overlap.

Minor comments

(1) Af1521 Gly42Glu mutant that eliminates AR binding should be shown in Figure 2.

Response. We described this mutant in a methods paper (Kamata et al., *Methods in Molecular Biology*, 2019), which is now cited in the text.

(2) The blots that have numbers on the left side should make is clear what the numbers represent (e.g., Figures 1A and 3K).

Response. We improved the panel spacing, labels, and the figure legends.

(3) Exposure of some blots make it difficult to see if there is a band present (e.g., Figures 2D and 4E).

Response. We have tried to make sure all blots and figures accurately reflect the results. All detection was done using fluorescently labeled antibodies and IR scanning, which though imperfect, is sensitive and linear for the antibodies we have tested. All gels and blots are also contained in the Source Data.

(4) The possibility of Parp9 macrodomain hydrolase activity was not discussed.

Response. We agree this is an interesting idea, but we did not observe an effect on AR ADP-ribosylation levels when comparing WT to Parp9 knockout cells. Having said this, we did not comment on this in the manuscript because we feel a dedicated experiment would be necessary to address the question.

REVIEWERS' COMMENTS

Reviewer #1 (Remarks to the Author):

I thank the authors for their revisions, which have adequately addressed most of my previous comments.

The new experiment in Fig 6I-K convincingly addresses my biggest concern.

The only remaining concern I have is how is this complex modulating AR activity as opposed to free-AR. Initially I thought that the idea was that ADPr AR has increased binding to the target gene, but according to the data and their response this does not seem to be the case. This was also raised by rev2 and I am not sure if the authors have addressed this point very convincingly: the PSA experiment doesn't inform on how the complex affects expression.

Reviewer #3 (Remarks to the Author):

The authors have done a good job of addressing my concerns. I am satisfied. This paper provides new information about the AR signaling pathway, adding a new level of understanding and connectivity between ADP-ribosylation and steroid hormone signaling.

REVIEWERS' COMMENTS

Reviewer #1 (Remarks to the Author):

I thank the authors for their revisions, which have adequately addressed most of my previous comments.

The new experiment in Fig 6I-K convincingly addresses my biggest concern.

The only remaining concern I have is how is this complex modulating AR activity as opposed to free-AR. Initially I thought that the idea was that ADPr AR has increased binding to the target gene, but according to the data and their response this does not seem to be the case. This was also raised by rev2 and I am not sure if the authors have addressed this point very convincingly: the PSA experiment doesn't inform on how the complex affects expression.

Response

We added following to the discussion (page 15): Exactly how Dtx3L/Parp9 promotes transcription remains to be determined. One scenario is that it involves the E3 ligase activity of Dtx3L, which is known to modify histones, but an effect on co-factor interactions with AR is also possible. AR binding to chromatin appears to be independent of ADP-ribosylation, based on our finding that Olaparib inhibition of AR ADP-ribosylation did not detectably reduce AR binding to the MYBPC1 promoter.

Reviewer #3 (Remarks to the Author):

The authors have done a good job of addressing my concerns. I am satisfied. This paper provides new information about the AR signaling pathway, adding a new level of understanding and connectivity between ADP-ribosylation and steroid hormone signaling.